# Research Progress on the Functional Regulation Mechanisms of ZKSCAN3

**DOI:** 10.3390/biom15071016

**Published:** 2025-07-14

**Authors:** Jianxiong Xu, Xinzhe Li, Jingjing Xia, Wenfang Li, Zhengding Su

**Affiliations:** 1School of Pharmaceutical Science, Institute of Materia Medica, Xinjiang University, Urumqi 830017, China; jianxiongxu@stu.xju.edu.cn (J.X.); xiajing94@xju.edu.cn (J.X.); 2College of Life Science and Technology, Xinjiang University, Urumqi 830017, China; lixinzhe@stu.xju.edu.cn

**Keywords:** ZKSCAN3, functional regulation, mechanism, signal pathways

## Abstract

The zinc finger protein with KRAB and SCAN domains 3 (ZKSCAN3) has emerged as a critical regulator of diverse cellular processes, including autophagy, cell cycle progression, and tumorigenesis. Structurally, ZKSCAN3 is characterized by its conserved DNA-binding zinc finger motifs, a SCAN domain mediating protein–protein interaction, and a KRAB repression domain implicated in transcriptional regulation. Post-translational modifications, such as phosphorylation and ubiquitination, dynamically modulate its subcellular localization and activity, enabling context-dependent functional plasticity. Functionally, ZKSCAN3 acts as a master switch in autophagy by repressing the transcription of autophagy-related genes under nutrient-replete conditions, while its nuclear-cytoplasmic shuttling under stress conditions links metabolic reprogramming to cellular survival. Emerging evidence also underscores its paradoxical roles in cancer: it suppresses tumor initiation by maintaining genomic stability yet promotes metastasis through epithelial–mesenchymal transition induction. Furthermore, epigenetic mechanisms, including promoter methylation and non-coding RNA regulation, fine-tune ZKSCAN3 expression, contributing to tissue-specific outcomes. Despite these insights, gaps remain in understanding the structural determinants governing its interaction with chromatin-remodeling complexes and the therapeutic potential of targeting ZKSCAN3 in diseases. Future investigations should prioritize integrating multi-omics approaches to unravel context-specific regulatory networks and explore small-molecule modulators for translational applications. This comprehensive analysis provides a framework for advancing our mechanistic understanding of ZKSCAN3 and its implications in human health and disease. This review synthesizes recent advances in elucidating the regulatory networks and functional complexity of ZKSCAN3, highlighting its dual roles in physiological and pathological contexts.

## 1. Introduction

ZKSCAN3 has garnered increasing attention as a multifunctional regulator at the intersection of cellular homeostasis and disease pathogenesis [1]. As a member of the Krüppel-associated box (KRAB)-containing zinc finger protein family, ZKSCAN3 integrates DNA-binding activity with protein–protein interaction capabilities, enabling it to modulate transcriptional networks that govern fundamental cellular processes [2,3,4,5]. Initially identified as a transcriptional repressor, its biological significance has expanded to encompass roles in autophagy regulation, genomic stability maintenance, and metabolic adaptation, underscoring its importance in both physiological and pathological contexts [6]. ZKSCAN3 is defined by three functional domains: an N-terminal SCAN domain facilitating oligomerization and interaction with co-regulators, a central KRAB repression domain mediating transcriptional silencing, and a C-terminal array of zinc finger motifs critical for DNA sequence recognition [7,8,9,10,11,12]. This modular architecture allows ZKSCAN3 to dynamically engage chromatin-remodeling complexes and epigenetic modifiers, thereby influencing gene expression profiles in a context-dependent manner [5,6]. Although the direct structure of ZKSCAN3 has not been resolved, homologous protein studies of the SCAN/KRAB structural domains suggest that such domains can mediate protein interactions through dimerization or recruit chromatin-modifying complexes (e.g., KRAB-KAP1) to achieve transcriptional repression [8,13,14]. These insights highlight ZKSCAN3 as an exemplification for understanding how protein domain organization translates into functional versatility. Functionally, ZKSCAN3 emerges as a nodal point in cellular stress responses. Under nutrient-rich conditions, it acts as a master repressor of autophagy by directly binding to the promoter regions of autophagy-related genes (ATGs), thereby suppressing their transcription. This regulatory axis is disrupted during starvation or metabolic stress, where ZKSCAN3 undergoes cytoplasmic relocation, relieving transcriptional repression and enabling autophagic flux [6,15,16]. Such plasticity positions ZKSCAN3 not merely as a passive responder but as an active orchestrator of metabolic reprogramming [17]. Concurrently, its role in maintaining genomic integrity through regulation of DNA damage repair pathways and telomere homeostasis further underscores its significance in preserving cellular fitness. Paradoxically, ZKSCAN3 exhibits context-dependent oncogenic or tumor-suppressive activities. In early tumorigenesis, it suppresses malignant transformation by inhibiting epithelial–mesenchymal transition (EMT) and limiting genomic instability [18]. However, in advanced cancers, its overexpression correlates with metastatic progression and therapeutic resistance, potentially through reprogramming of metabolic dependencies or modulation of tumor microenvironment interactions. This duality complicates therapeutic targeting efforts but also hints at tissue-specific opportunities for intervention [9,18].

Despite these advances, critical gaps persist in understanding the molecular mechanisms governing ZKSCAN3’s functional dichotomy [5]. How do post-translational modifications—such as ubiquitination or acetylation—fine-tune its interaction with chromatin modifiers? To what extent do non-coding RNAs or epigenetic modifiers contribute to its tissue-specific expression patterns? Furthermore, the mechanism of conformational change by which ZKSCAN3 switches between transcriptional repression and activation remains unclear.

This review synthesizes current knowledge on ZKSCAN3’s regulatory networks and functional plasticity, emphasizing emerging themes in autophagy, cancer biology, and metabolic regulation. By integrating insights from biochemical, genomic, and clinical studies, we aim to provide a cohesive framework for understanding ZKSCAN3’s biological complexity and to identify unexplored avenues for therapeutic exploitation.

## 2. Gene Structure and Transcript Variants of ZKSCAN3

The *ZKSCAN3* is localized on chromosome 6 (6p22. 1) and encodes a member of the C_2_H_2_-type zinc-finger transcription factor family containing KRAB and SCAN structural domains [19,20]. Among its protein function modules, the KRAB structural domain plays a pro- or oncogenic role in biological processes such as cell proliferation, apoptosis, migration, and invasion through epigenetic regulation, transcriptional repression or activation, and interactions with other proteins [7,8,9]. The KRAB domain is a modular protein segment comprising approximately 75 to 100 amino acids [20,21]. Its core consists of a highly conserved approximately 45-amino acid A-box and a less conserved approximately 30-amino acid B-box. The A-box forms two stable α-helices whose hydrophobic core maintains structural stability, while surface-exposed acidic residues play key roles in mediating protein–protein interaction. The B-box indirectly supports overall function by enhancing the activity of the A-box [20,21]. The KRAB domain contains intrinsically disordered regions at its N-terminal and C-terminal ends to confer conformational flexibility, while its core helical topology is highly conserved across mammals, reflecting its critical functional importance [8,21]. In KRAB zinc finger proteins (KRAB-ZFPs), the KRAB-A box interacts with the RBCC domain (comprising RING, B-box, and coiled-coil regions) of KAP1 via hydrophobic interactions [14]. Upon binding, KAP1 dimerizes and recruits chromatin remodeling complexes such as HP1, SETDB1, and NuRD, which modify chromatin through H3K9 trimethylation (H3K9me3) and deacetylation to repress target gene transcription. Furthermore, the RBCC domain of KAP1 facilitates the nuclear import of KRAB-ZFPs through a “piggy-back” mechanism, enabling their function in transcriptional regulation [8,14,20,21,22]. The SCAN structural domain is relatively conserved and is involved in the regulation of gene expression and cell differentiation through homo-oligomerization or heterodimerization, but its precise molecular mechanism remains to be studied [10,11,12]. Another study found that ZKSCAN3 dynamically regulates replication fork stability through the SCAN structural domain, and its function is genetic background-dependent. Breast cancer susceptibility gene 1 (BRCA1) is a key tumor suppressor gene [23,24]. In BRCA1-proficient cells, ZKSCAN3 synergistically inhibits nuclease Exo1/Mre11-mediated fork over-resection with BRCA1 to maintain chromosome stability. In contrast, in BRCA1-deficient cells, ZKSCAN3 reverses its function and promotes resection through the SCAN structural domain, exacerbating genomic abnormalities. This mutually exclusive relationship reveals the dual role of ZKSCAN3 as both a fork protection factor and a promoter of resection under specific conditions [25].

The *ZKSCAN3* produces three mRNA transcript variants by variable splicing. Variant 1 contains the complete SCAN, KRAB, and zinc finger structural domains and translationally generates the longer protein isoform 1 [1,5]. KRAB structural domains, by recruiting co-repressors (e.g., KAP1), form a chromatin-modifying complex (SETDB1/HP1), which in turn represses autophagy genes (e.g., Microtubule-associated protein 1A/1B-light chain 3B (*LC3b*) and transposon activity [26,27,28]. In the tumor environment, variant 1 promotes cell proliferation, migration, and angiogenesis by activating the integrin β4 (ITGβ4)/vascular endothelial growth factor (VEGF) pathway. Variant 2 is unable to bind to KAP1 due to the absence of the KRAB structural domain, but its coding region partially overlaps with that of variant 1 and can encode the same protein as variant 1 [26]. In normal cells, variant 2 depends on the SCAN structural domain to mediate protein interactions and maintain cellular homeostasis. In tumors, it binds to pro-oncogenic factors such as β-catenin to enhance cell invasiveness. The zinc finger region of variant 3 is altered by splicing, resulting in a shift in DNA binding specificity, and differences in its 5′UTR and the use of downstream start codons allow it to encode an N-terminally truncated protein isoform 2. In tumors, variant 3 targets genes such as CyclinD2 (CCND2) and drives aberrant tumor cell proliferation. In normal cells, it is involved in the process of differentiation or apoptosis regulation (Figure 1) [1,10,11].

## 3. ZKSCAN3 Exerts Multifunctional Regulatory Mechanisms in Normal Physiological Processes

As a multifunctional regulator, ZKSCAN3 plays a key role in physiological processes such as stem cell homeostasis, erythropoiesis, cardiac remodeling, autophagy regulation, and metabolic homeostasis by directly or indirectly regulating gene expression, epigenetic modifications, and cellular signaling pathways. However, its function shows significant variability in different cell types, suggesting the complexity and environment-dependence of its mechanism of action (Figure 2).

Stem cell failure, as a central mechanism of decreased regenerative capacity of aging-related tissues, is closely associated with the development of age-related diseases [29]. ZKSCAN3 was found to exert an autophagy-independent senescence inhibition function in human mesenchymal stem cells (hMSCs) through an epigenetic regulatory network [30]. In senescent hMSCs, ZKSCAN3 expression is significantly down-regulated, and its deletion leads to the loss of heterochromatin stability, which is manifested by the aberrant dissociation of nuclear lamina-associated domains (LADs) and uncontrolled transcription of repetitive sequences, and ultimately accelerates the process of cellular senescence [31]. Mechanistic studies have shown that ZKSCAN3 maintains chromatin high-level structure by anchoring LADs and promotes the deposition of repressive histone modifications (e.g., H3K9me3), thereby preventing the aberrant activation of transposons (LINE-1 and ERVs) triggered by DNA derepression in heterochromatin regions [32]. This epigenetic regulation not only inhibits genomic instability but also blocks aging-associated cGAS-STING pathway activation and SASP inflammatory factor secretion [33]. Notably, this mechanism is also present in fibroblast and mouse stem cell models [33]. Functional replication experiments confirmed that overexpression of ZKSCAN3 rescued the premature aging phenotype of defective hMSCs and restored the proliferative capacity, while CRISPR/Cas9-mediated knockout models revealed the molecular basis of its ability to retard aging by stabilizing the epigenetic landscape in human embryonic stem cells (hESCs) and hMSCs [31,34]. These findings provide new strategies for targeted intervention in stem cell senescence. Notably, the functional diversity of ZKSCAN3 is not limited to the regulation of stem cell senescence, and its role in the blood system is gradually being revealed.

It was found that ZKSCAN3 regulates key transcription factors for erythroid development through a dual mechanism: on the one hand, it directly binds to GATA1 and inhibits its transcriptional activity, and on the other hand, it promotes the expression of KLF1, which maintains erythropoietic homeostasis through the modulation of erythroid lineage differentiation marker genes (e.g., genes related to hemoglobin synthesis), respectively [35,36,37]. This regulatory mechanism of ZKSCAN3 significantly inhibits erythropoiesis. In the ZKSCAN3 knockout mouse model, the number of splenic red lineage progenitors (early) and mature erythrocytes (late) is abnormally increased, while bone marrow late erythrocytes are significantly reduced, ultimately triggering hemolytic anemia [35]. Further mechanistic studies showed that ZKSCAN3 also binds to the promoter region of the *Tiam1* gene and represses its expression, whereas the abnormal expression of Tiam1, a key regulator of the DNA damage response, may exacerbate the anemia process by interfering with the erythrocyte apoptotic program [35,38,39]. These findings not only reveal the molecular basis of ZKSCAN3 in maintaining erythrocyte dynamic homeostasis through the regulation of the GATA1-KLF1 axis and apoptosis-related pathways but also provide a new strategy for the targeted treatment of hemolytic anemia and other blood diseases.

The regulatory function of ZKSCAN3 cuts across multiple physiological processes, from stem cells to terminally differentiated cells. In cardiomyocytes, it maintains cardiac homeostasis by dynamically regulating the transcriptional network of autophagy-related genes and cardiac remodeling-related target genes. Targeting ZKSCAN3 or its downstream pathway may be a potential strategy to restore autophagy homeostasis and treat cardiac diseases [40]. It was found that cardiomyocyte-specific knockdown of ZKSCAN3 disrupted the balance between autophagy activation and inhibition and significantly exacerbated pathological cardiac remodeling induced by transverse aortic constriction (TAC) [41]. Experiments showed that knockout mice exhibited higher mortality, cardiac hypertrophy index, and decreased cardiac function (e.g., reduced ejection fraction) compared with controls. In terms of molecular mechanisms, ZKSCAN3 deletion led to a significant down-regulation of the expression levels of autophagy genes (e.g., *LC3*), whereas TAC surgery had an inhibitory effect on autophagy genes in control mice but no superimposed effect on knockout mice, suggesting that ZKSCAN3 is a central hub for autophagy regulation under cardiac stress conditions [42]. This evidence highlights the critical role of ZKSCAN3 in maintaining cardiac physiological function and inhibiting disease progression.

ZKSCAN3 maintains cellular homeostasis through a tissue-specific transcriptional regulatory network, a property that is highly conserved in a variety of terminally differentiated cells. For example, in retinal pigment epithelial (REP) cells, ZKSCAN3 regulates the autophagy-lysosome system through dynamic nucleoplasmic shuttling. ZKSCAN3 represses autophagy- and lysosome-related genes, such as *DIRAS3*, *LC3B*, and Cathepsin B. It localizes to the nucleus to exert transcriptional repression during nutrient sufficiency, whereas under nutrient stress conditions, nuclear export mediated through the p38 MAPK signaling pathway deregulates autophagy genes to activate autophagic flow and enhance lysosomal function [43]. This specific mode of regulation has shown significant homology in evolution [19]. Drosophila modeling studies have shown that ZKSCAN3 in the fat body represses autophagy-related genes by binding to the Motif 1 region of the genome, and its loss of function leads to aberrant autophagy activation in Motif 1-binding protein (M1BP)-deficient Drosophila [44]. In the Drosophila eye, M1BP affects eye development by regulating the Wingless (Wg) signaling pathway. Down-regulation of M1BP leads to aberrant induction of Wg expression, which inhibits eye formation [45]. ZKSCAN3 and M1BP not only share genomic target sites (e.g., metabolism-related gene promoter regions) but also control transcriptional processes through the modulation of the RNA polymerase II (Pol II) pause-release mechanism to control the transcription process. In vertebrate cells, heterologous expression of M1BP antagonizes ZKSCAN3-mediated inhibition of starvation-induced autophagy, confirming that the two are functionally homologous proteins [46]. ZKSCAN3 affects apoptosis and autophagy, the two mechanisms of cell death, by regulating the JNK signaling pathway through the transcriptional pausing mechanism [45]. This cross-species conservation not only reveals the centrality of the ZKSCAN3-M1BP axis in the regulation of metabolism and autophagy, but also the correlation between its dysfunction and human tumorigenesis suggests that the Drosophila model can be used as an important tool to study its oncogenic mechanism [46].

In addition to the above cells, it was found that ZFP407 interacts with the PPARγ/RXRα protein complex in the nucleus of adipocytes, and its binding sites significantly overlap with those of PPARγ in the genome, and these overlapping sites are enriched with transcription factor binding motifs such as CTCF, RARα/RXRγ, TP73, and ELK1, which are known to have a regulatory role in adipocyte development and function [47,48]. ZFP407 is associated with ZKSCAN3 binding sites in certain regions of the genome, suggesting that ZKSCAN3 may be involved in the non-PPARγ-dependent function of ZFP407 [49], thereby regulating adipocyte function. In addition, ZKSCAN3 was found to maintain immune homeostasis by limiting plasma cell overdifferentiation through inhibition of X-inactive-specific transcript (Xist) [50].

These functions demonstrate the complexity and diversity of ZKSCAN3 in normal cells, where it plays different roles in different cell types and tissues. With further research, the biological functions of ZKSCAN3 in normal cells will also be further elucidated, laying the foundation for uncovering its role in diseases.

## 4. The Function and Regulatory Mechanisms of ZKSCAN3 in Tumors

The pro-carcinogenic function of ZKSCAN3 was first revealed by Yang et al. [19] in 2008, and its pro-proliferative properties were demonstrated by in vitro and in vivo experiments in highly expressed colorectal cancer tissues. Overexpression significantly promoted tumor growth, while gene silencing induced cell cycle arrest and activated senescence-related phenotypes [6]. Follow-up studies have progressively revealed the extensive pro-carcinogenic effects of ZKSCAN3 in a variety of solid tumors (hepatocellular carcinoma, gastric cancer, breast cancer, cervical cancer, prostate cancer, bladder cancer, epithelial ovarian cancer) and hematological tumors (multiple myeloma) (Figure 3) [5]. These findings establish the status of ZKSCAN3 as a key driver of tumor progression and provide a molecular biological basis for the development of targeted intervention strategies [51].

### 4.1. ZKSCAN3 Drives Colorectal Cancer Progression by Activating Wnt/β-Catenin and ITGβ4/FAK/AKT Pathways

ZKSCAN3 plays a regulatory role in colorectal carcinogenesis. It regulates the expression of 280 target genes (204 up-regulated and 76 down-regulated) through specific DNA recognition motifs (KRDGGG), several of which are closely related to cell proliferation and survival [52,53]. Research indicates that the WNT/β-catenin pathway induces ZKSCAN3 gene expression via the β-catenin/TCF4 complex, with ZKSCAN3 levels closely associated with the presence of β-catenin within the nucleus, thereby establishing a reinforcing cycle that promotes tumor progression [54,55]. In invasive phenotypic cells, ZKSCAN3 activates ITGβ4/Focal Adhesion Kinase (FAK)/AKT cascade signaling by up-regulating key effector molecules such as VEGF and ITGβ4 [56]. Specifically, FAK phosphorylation triggers AKT signaling to drive aberrant cell proliferation and inhibit apoptosis, while promoting epithelial–mesenchymal transition (EMT) to enhance metastatic potential [52,54,57]. Analysis of clinical data from 450 colorectal cancer patients revealed that elevated ZKSCAN3 expression demonstrated a strong positive association with hepatic metastasis risk and higher serum carcinoembryonic antigen (CEA) concentrations. This protein’s abundance was markedly increased in subgroups exhibiting high CEA levels [57,58]. In addition, cancer-type OATP1B3 (Ct-OATP1B3), a tumor-specific isoform protein produced by selective splicing of the SLCO1B3 gene, is highly expressed in tissues such as colon cancer, while it is hardly expressed in normal tissues. Localized in the lysosomal membrane, it mediates chemoresistance by transporting drugs (e.g., encorafenib) to the lysosome to reduce intracellular concentrations [59]. Mutation of the ZKSCAN3 binding site in the promoter caused the activity of colon cancer cells to plummet (to 29.9% and 14.3% in DLD1 and T84 cell lines, respectively), whereas the activity of hepatocytes only decreased to 71.6% [54]. The EMSA assay confirmed that ZKSCAN3 directly binds to the promoter. The results revealed that ZKSCAN3 is a core transcription factor driving the specific expression of CtOATP1B3 colon cancer [60]. ZKSCAN3 overexpression also increased 5-fluorouracil (5-FU) resistance by 2.5-fold and maintained high expression in tumors with wild-type *APC*, *p53*, and *KRAS* genes [19,61,62]. This multidimensional evidence not only elucidates the mechanism by which ZKSCAN3 interacts with signaling pathways to drive malignant progression through the transcriptional regulatory network but also provides a theoretical basis for the development of precise therapeutic strategies targeting ZKSCAN3. In addition to its involvement in classical signaling pathway interactions, ZKSCAN3 forms a cross-level oncogenic network through deep coupling with epigenetic regulation and cellular metabolic homeostasis.

### 4.2. ZKSCAN3 Promotes Hepatocellular Carcinoma Progression via the FAK/AKT-Autophagy Inhibition Axis

The multidimensional regulatory mechanisms of ZKSCAN3 in hepatocellular carcinoma are gradually being revealed. In EMT regulation, it drives the EMT process by activating the FAK/AKT signaling axis through direct binding to the promoter of the *ITGβ4* [52,58], as evidenced by the up-regulation of the expression of E-cadherin and significant down-regulation of the mesenchymal markers N-cadherin, vimentin, and matrix metalloproteinases (MMPs) [9,58]. At the level of epigenetic regulation, the microRNA miR-124 antagonizes the promotion of EMT by targeting and inhibiting ZKSCAN3 expression, thereby maintaining epithelial cell integrity and inhibiting hepatocellular carcinoma metastasis [7,63,64]. In terms of autophagy and metabolic regulation, ZKSCAN3 impedes autophagic flow and lysosomal generation by inhibiting the expression of autophagy core genes such as *ULK1*, *LC3b*, *DFCP1,* and *WIPI2*, and silencing of ZKSCAN3 rescued this inhibition and activated cellular autophagic programs [17,65,66]. In addition, the oncogenic protein CHD1L inhibits the transcriptional activity of ZKSCAN3 by binding to it and enhances the migration of hepatocellular carcinoma cells by promoting the degradation of focal adhesion protein (Paxillin) and regulating focal adhesion (FA) dynamics [67]. Clinical therapeutic relevance studies have shown that ZKSCAN3 reduces the sensitivity of hepatocellular carcinoma cells to 5-FU chemotherapy by inhibiting their senescence, and its high expression is positively correlated with hepatocellular carcinoma proliferation, invasion, and lung metastatic potential. In conclusion, ZKSCAN3 is a key driver of hepatocellular carcinoma progression through the integration of multiple mechanisms, including EMT signaling activation, epigenetic imbalance, autophagy inhibition, and chemoresistance, providing a multidimensional entry point for the development of targeted intervention strategies.

### 4.3. ZKSCAN3 Facilitates Gastric Cancer Progression Through Ras/MAPK-MST1R-MMP/VEGF Multi-Axis Signaling

The multilevel oncogenic mechanism of ZKSCAN3 in hepatocellular carcinoma provides important clues to analyze its functional commonality and specificity in other digestive tract tumors. This transcription factor also has a central regulatory role in gastric cancer, but its mode of action shows a unique epigenetic modification dependence. The regulatory role of ZKSCAN3 in gastric cancer and its pro-carcinogenic mechanism have been gradually clarified. Regarding epitranscriptomic control, ALKBH5 promotes the stability and translation of ZKSCAN3 transcripts by selectively erasing m^6^A modifications on ZKSCAN3 mRNA [68]. Knockdown of *ALKBH5* results in the accumulation of m^6^A modification and down-regulation of ZKSCAN3 mRNA expression, whereas overexpression of ALKBH5 significantly up-regulates ZKSCAN3 expression by decreasing the m^6^A levels [68,69]. Clinicopathologic analysis showed that 32.2% of gastric cancer samples had abnormally high expression of ZKSCAN3, and the intensity of its expression was significantly and positively correlated with lymphatic metastasis and distant dissemination. In terms of molecular mechanisms, ZKSCAN3 drives malignant progression through multiple pathways: (1) up-regulation of Mitogen-Activated Protein Kinase Kinase 2 (MEK2), RAS Guanyl Releasing Protein 2 (RasGRP2), Insulin-Like Growth Factor 2 (IGF-2), and ITGβ4 to promote tumor proliferation; (2) activation of the MST1R signaling axis and inducing the expression of MMP-2/9 and cathepsin D to enhance invasive metastasis; and (3) stimulation of VEGF secretion to promote tumor angiogenesis [69,70,71]. Prognostic correlation studies have shown that the overall survival (OS) of ZKSCAN3-positive gastric cancer patients was significantly shorter than that of negative patients, indicating its potential application as a biomarker and therapeutic target for evaluating prognosis in gastric cancer [71]. In conclusion, ZKSCAN3 is a key driver of gastric cancer progression through the integration of epigenetic regulation, activation of multiple signaling pathways, and microenvironmental remodeling, and the epigenetic and multipathway synergistic regulation model established by ZKSCAN3 in gastric cancer provides key clues for exploring its pan-cancer oncogenic mechanism. Its pro-carcinogenic role is further expanded in breast cancer.

### 4.4. ZKSCAN3 Accelerates Breast Cancer Proliferation and Invasion via AKT/mTOR-Cyclin D1-MMP Axis

ZKSCAN3 exhibits significant pro-oncogenic properties in breast cancer, and its high expression synergistically drives malignant progression through multiple signaling pathways [72]. Studies have shown that ZKSCAN3 promotes the proliferation, migration, and invasive ability of breast cancer cells (e.g., MCF-7, MDA-MB-231) by regulating cell cycle and apoptosis-related molecular networks [72,73,74]. At the mechanistic level, short hairpin RNA (shRNA)-mediated knockdown of ZKSCAN3 significantly inhibited the above phenotypes, as evidenced by the down-regulation of cyclin D1 (CCND1), B-cell lymphoma/leukemia 2 (BCL-2), and MMP-2/9 expression, while the expression of pro-apoptotic factor BCL-2 associated X protein (BAX) was up-regulated [9,63,72]. Animal models further verified that *ZKSCAN3*-shRNA significantly inhibited graft tumor growth by suppressing the AKT/mTOR signaling axis [6,58,75]. Notably, ZKSCAN3 also induced cell death by activating the BCL-2 family-dependent external apoptotic pathway in bladder and breast cancers, suggesting its dual regulatory functions in different cancer types [72,74]. Although the molecular details of its action network still need to be deeply analyzed, the available evidence has highlighted the potential value of ZKSCAN3 as a therapeutic target in breast cancer.

### 4.5. ZKSCAN3 Drives Cervical Cancer Proliferation and Metastasis Through RAS-MAPK/MST1R/VEGF Pathway

The dual regulatory properties and synergistic effects across signaling pathways exhibited by ZKSCAN3 in breast cancer provide important insights to reveal its functional diversity in other solid tumors. In cervical cancer, the oncogenic effect of ZKSCAN3 was found to be further realized through genomic copy number amplification and multidimensional transcriptional reprogramming. Lee et al. [76] found that genomic analysis revealed that the gene copy number amplification of ZKSCAN3 in cervical cancer tissues led to a significantly higher expression of its protein than that in normal tissues adjacent to the cancer. Clinical studies further showed that high expression of ZKSCAN3 was significantly associated with shorter OS and higher risk of disease recurrence in patients, and its high expression in patients with low-grade cervical cancer was associated with worse progression-free survival (PFS) and OS, suggesting that it could be used as an independent prognostic indicator [5,76]. At the level of molecular mechanisms, ZKSCAN3 drives the malignant phenotype through a multi-targeted transcriptional regulatory network: (1) transcriptional activation of MEK2, RasGRP2, IGF-2, and ITGβ4 (key effector molecules) promotes proliferation; (2) activation of the MST1R signaling axis enhances migration; (3) induces VEGF-mediated tumor angiogenesis; and (4) up-regulates matrix metalloproteinase 26 (MMP26), cathepsin D, and protein hydrolases such as serine protease 3 (PRSS3) to enhance invasive potential [76]. These results not only clarified how ZKSCAN3 facilitates cervical cancer advancement through the coordination of proliferation, metastasis, and alterations in the tumor microenvironment, but also offered experimental support for its use as a prognostic biomarker and potential therapeutic target [77].

### 4.6. ZKSCAN3 Promotes Prostate Cancer Proliferation and Metastasis via VEGF/ITGβ4-Cyclin D-NFκB-MMP Axis

The genomic amplification-driven and multi-targeted transcriptional regulation pattern of ZKSCAN3 revealed in cervical cancer provides new perspectives to explore its cross-organ oncogenic mechanism in prostate cancer. ZKSCAN3 drives the malignant phenotype through a multifaceted regulatory network in prostate cancer. Functional studies have shown that its high expression in PC3 cells significantly enhances migration, invasion, and loss of cell adhesion, whereas gene silencing of ZKSCAN3 inhibits colony formation, reduces cell viability, and induces apoptosis [74]. Molecularly, ZKSCAN3 promotes tumor proliferation and angiogenesis and enhances metastatic potential through transcriptional activation of the expression of key genes such as VEGF, ITGβ4, and CCND D1/D2. In addition, ZKSCAN3 drives tumor invasion through NF-κB and MMP pathways [78]. Clinicopathological analysis showed that the ZKSCAN3 expression level was significantly and positively correlated with aggressive pathological features of prostate cancer (e.g., perineural invasion) and the risk of postoperative recurrence, suggesting that it could be used as an independent prognostic marker [78]. In addition, epidemiological studies have suggested that ZKSCAN3 may be involved in the regulation of prostate cancer risk after vasectomy, but its specific mechanism still needs to be explored in depth [79,80,81]. It is noteworthy that ZKSCAN3 also exhibits cross-system biological diversity by regulating lysosomal function, autophagy activity, and respiratory homeostasis [82,83], which further highlights its role in tumor microenvironment modulation.

### 4.7. ZKSCAN3 Inhibits Malignant Progression of Pancreatic Cancer by Targeting ULK1/LC3-II Autophagy Axis

While the pro-invasive and microenvironmental remodeling mechanisms established by ZKSCAN3 in prostate cancer demonstrate its conserved function as a cancer driver, its unique mode of action in pancreatic cancer reveals the complexity of tissue-specific regulation. Nucleolin NOP53 was found to up-regulate the expression of ZKSCAN3 by enhancing its transcriptional activity or protein stability. Knockdown of *NOP53* resulted in a significant decrease in ZKSCAN3 mRNA and protein levels, accompanied by accumulation of the autophagy marker LC3-II [84]. Clinical correlation analysis showed that high ZKSCAN3 expression was positively correlated with prolonged OS in pancreatic cancer patients, a phenomenon that is contrary to its pro-carcinogenic role in most solid tumors, suggesting that its function may be affected by the tumor microenvironment or tissue-specific signal reprogramming [62,85]. Mechanistic studies have demonstrated that ZKSCAN3 inhibits pancreatic cancer cell proliferation, migration, and invasion by transcriptionally activating the expression of autophagy core genes, such as *ULK1* and LC3-II, and promoting autophagosome formation and lysosomal degradation functions [5,62]. In vivo experiments further confirmed that MIA PaCa-2 cells silencing ZKSCAN3 formed larger tumor sizes in a nude mouse model, accompanied by up-regulation of LC3-II expression, elevation of the Ki-67 proliferation index, and enhancement of autophagic flow, suggesting that it promotes tumor growth by inhibiting autophagy [62,71,86]. This functional paradox suggests that targeting the ZKSCAN3-autophagy regulatory axis may provide a novel intervention strategy for pancreatic cancer treatment.

### 4.8. ZKSCAN3 Activates Bladder Cancer Invasion and Proliferation via c-Myc/FGFR3-MMP2/9 Signaling Axis

The “autophagy-activated-oncogenic” paradox of ZKSCAN3 in pancreatic cancer challenged the classical perception of its unidirectional pro-carcinogenic function in solid tumors, whereas the mode of action of ZKSCAN3 in bladder cancer returned to the conserved characteristics of oncogenic factors [62]. In bladder cancer, the oncogenic function of ZKSCAN3 is similar to its mode of action in prostate cancer. By silencing *ZKSCAN3*, the colony formation ability, cell viability, and invasive migration properties of UMUC3 and 647V bladder cancer cells were significantly inhibited, which in turn blocked the tumor growth and metastatic process [84,87]. On a molecular level, silencing ZKSCAN3 decreased the levels of oncogenes c-Myc, FGFR3, and MMP2/MMP9, while increasing the expression of tumor suppressor genes p53 and PTEN, thereby altering the signaling pathways that promote apoptosis and inhibit cell proliferation [88,89]. In vivo experiments further confirmed that *ZKSCAN3*-shRNA-treated bladder cancer cells had significantly diminished tumorigenicity in a xenograft mouse model, as evidenced by a reduction in tumor size and malignancy [54,87]. These results not only clarify the molecular basis of ZKSCAN3 driving bladder cancer progression through multi-target regulation but also suggest that RNA interference technology targeting ZKSCAN3 has the potential to be translated into clinical therapeutic strategies.

### 4.9. ZKSCAN3 Synergizes with EGFR to Activate Pro-Survival, Anti-Apoptotic, and Tumor Microenvironment Remodeling Pathways in Ovarian Cancer

The independent pro-carcinogenic mechanism and therapeutic potential of RNA interference established by ZKSCAN3 in bladder cancer provide an important basis for exploring its interaction with other oncogenic signaling pathways. In epithelial ovarian cancer, ZKSCAN3 has shown a more complex oncogenic logic by forming a synergistic regulatory network with epidermal growth factor receptor (EGFR), which is a member of the receptor tyrosine kinase superfamily I [6]. EGFR, as a member of the receptor tyrosine kinase superfamily I, participates in the regulation of signaling pathways such as cell proliferation, survival, and migration through ligand-dependent activation, and its gene amplification and functional mutation are the most important factors in the regulation of cell proliferation, survival, and migration. Gene amplification and functional mutations are important mechanisms driving tumorigenesis [90,91]. In epithelial ovarian cancer, ZKSCAN3 and EGFR showed a synergistic pattern of high expression, and their expression levels in malignant tissues were significantly higher than those in normal ovarian tissues and benign tumor tissues, and the expression intensity showed a significant positive correlation [66]. This co-expression feature is closely related to the high lethality and treatment resistance of ovarian cancer. Given the potential synergistic role of the EGFR-ZKSCAN3 signaling axis in the remodeling of the tumor microenvironment, the combined detection of the expression levels of the two may provide new biomarker combinations for the early diagnosis, prognostic assessment, and the design of targeted therapeutic strategies for ovarian cancer, with potential clinical translational value [92,93].

### 4.10. ZKSCAN3 Enhances Cell Cycle Progression and Angiogenesis in Multiple Myeloma by Regulating CCND2 and VEGF

The synergistic oncogenic network formed by ZKSCAN3 with EGFR in ovarian cancer reveals a unique mechanism for remodeling the microenvironment through receptor tyrosine kinase signaling in solid tumors, and its functional studies in multiple myeloma further broaden its scope of action. In multiple myeloma, ZKSCAN3 demonstrates important functional characteristics [63]. Its expression level is significantly elevated in myeloma tissues compared with normal samples, suggesting its potential oncogenic role [9,52]. Mechanistic studies revealed that ZKSCAN3 directly activates the transcriptional activity of CCND2 and VEGF by specifically binding to their promoter regions. Overexpression of ZKSCAN3 enhances CCND2 promoter activity and induces protein expression [94], whereas gene silencing significantly reduces CCND2 levels, which in turn inhibits myeloma cell proliferation [17,63]. This targeted regulation pattern was further confirmed by functional response assays: the pro-proliferative effect of ZKSCAN3 was positively correlated with the CCND2 expression level [95]. Together, this evidence reveals a central mechanism by which ZKSCAN3 drives multiple myeloma progression through transcriptional regulation of the CCND2-VEGF signaling axis.

ZKSCAN3 plays an important role in tumor progression by regulating key molecules such as VEGF, ITGβ4, MMP2, MMP9, CCND1, and CCND2 [5]. Its expression level is now found to be significantly elevated in a variety of cancers and positively correlated with poor patient prognosis. Mechanistic studies have shown that ZKSCAN3 exhibits dual potential as a diagnostic marker and therapeutic target by regulating malignant biological behaviors such as cell proliferation, invasive migration, apoptosis inhibition, and autophagy homeostasis (Table 1) [6,54,72]; however, its tissue-specific expression pattern and upstream/downstream regulation in various types of tumors still need to be investigated in depth.

## 5. The Pathological Regulatory Role and Therapeutic Potential of ZKSCAN3 in Multiple Diseases

ZKSCAN3, as a multifaceted regulatory factor, is not only involved in the regulation of physiological functions of a wide range of normal cells, but also widely mediates the pathologic processes of various diseases such as Alzheimer’s disease (AD), Parkinson’s disease (PD), type 2 diabetes mellitus-related atherosclerosis (T2DM-AS), spinal bulbar muscular atrophy (SBMA), acute pancreatitis (AP), acute lung injury (ALI), and sepsis (Figure 4).

In AD models, down-regulation of *ZKSCAN3* expression leads to significant up-regulation of the expression of its downstream lysosome-related genes (e.g., *GRB2, Lamp2*), which promotes lysosomal generation and enhanced function [96,97]. Specifically, in the AD model group, both *ZKSCAN3* gene and protein expression were significantly reduced, accompanied by enhanced lysosomal activity, reduced neuronal damage, and improved synaptic structure [98]. In the *ZKSCAN3* silencing experiment, the expression of genes related to lysosomal function was further elevated, and the lysosomal number and activity were increased, confirming that ZKSCAN3 is a key negative regulator of lysosomal function [99]. In addition, the combined intervention of chitosanase 1 (CHIT1) and calcitonin gene-related peptide (CGRP) significantly inhibited ZKSCAN3 expression, which in turn enhanced lysosomal function and reduced neuronal apoptosis caused by pathologic protein Aβ toxicity [100,101]. This offers a possible approach for managing AD [102].

Autophagy dysfunction exacerbates dopamine neuron damage in the MPTP/MPP^+^-induced PD model. Sirtuin 1 (SIRT1) enhances autophagic flow and reduces neuronal damage by modifying ZKSCAN3 through deacetylation, inducing its extra-nuclear translocation, and relieving its inhibitory effect on autophagy genes, a process that relies on SIRT1 activation regulated by ROS levels [103,104,105]. However, some studies have found that knockdown of ZKSCAN3 failed to significantly attenuate α-synuclein aggregation or neuronal degeneration despite restoring lysosomal function [106], suggesting that the mechanism of action needs to be further elucidated. ZKSCAN3 exhibits an antagonistic effect with transcription factor EB (TFEB) in the regulation of autophagy [7,107]. ZKSCAN3, through the inhibition of TFEB nuclear translocation, blocks expression of lysosomal biosynthesis genes (e.g., *Lamp1*), leading to blockage of the autophagy-lysosomal pathway, whereas interference with ZKSCAN3 enhances TFEB activity and promotes α-synuclein clearance [108,109,110]. Notably, leucine repeat kinase 2 (LRRK2) mutations activate TFEB through a pathway independent of mTOR-ZKSCAN3, but its regulation is still closely linked to ZKSCAN3-mediated lysosomal function [111,112,113,114]. In summary, targeting the SIRT1-ZKSCAN3 axis or TFEB activation, combined with lysosomal-enhancing compounds (e.g., HEP14, LH2-051) and autophagy-targeting technologies (ATTEC/AUTAC), may provide multiple strategies for PD treatment, but the precise regulatory mechanisms of ZKSCAN3 need to be explored in depth [115]. Studies have revealed that the A30P mutant α-synuclein suppresses the JNK signaling pathway, reduces ZKSCAN3 phosphorylation, and promotes its nuclear accumulation, ultimately leading to impaired autophagy flux in midbrain dopaminergic neurons [116]. The molecular mechanisms of autophagy regulation have therapeutic potential not only in neurodegenerative diseases such as AD and PD, but also in diabetes-related cardiovascular complications.

In a type 2 diabetes mellitus (T2DM)-related atherosclerosis (AS) model, Nε-carboxyethyl lysine (CEL) inhibited autophagy-related gene expression by promoting the nuclear localization of ZKSCAN3, leading to impaired macrophage autophagy and decreased plaque stability [117]. Specifically, CEL inhibited macrophage autophagy and exacerbated plaque instability by regulating ZKSCAN3 acetylation through the RAGE/LKB1/AMPK1/SIRT1 pathway [117]. The resulting modulation of this signaling pathway is expected to develop innovative therapeutic avenues for the prevention and treatment of diabetes-related cardiovascular complications.

In addition, ZKSCAN3 also exhibits bidirectional regulation in neuromuscular diseases such as SBMA. Autophagy activation was demonstrated to be a key pathological feature in its cellular model and knock-in mice in SBMA studies. Further screening revealed that plant-derived small molecule compounds HEP14 and HEP15 could significantly promote lysosomal generation by inducing TFEB nuclear translocation and inhibiting ZKSCAN3 activity. Mechanistic studies have shown that such compounds mediate the phosphorylation inactivation of ZKSCAN3 through activation of protein kinase C (PKC), dependent on the JNK/p38 MAPK signaling pathway [15,118,119]. Consequently, selective regulation of the TFEB-ZKSCAN3 gene regulatory network could emerge as a promising therapeutic avenue in SBMA management [118]. In addition to this, in renal diseases (e.g., cystic kidney disease, acute kidney injury) with a similar mechanism, inactivation of ZKSCAN3 improves cellular clearance by enhancing autophagic flux, which serves as a key regulatory hub to alleviate the disease [120]. The study of autophagy-related transcriptional regulatory networks has been extended from neurodegenerative diseases to the field of inflammatory diseases, offering certain possibilities for cross-disease therapy.

In AP, an imbalance in the transcriptional regulation of autophagy exacerbates the pathological process through the synergistic effect of epigenetic modification and ZKSCAN3. As one of the key regulators of autophagy, aberrant expression of ZKSCAN3 has been found to be closely associated with AP progression [31,121]. Studies have shown that AP, similar to gastric cancer, suffers from an AlkB homolog 5 (ALKBH5)-mediated demethylation mechanism of ZKSCAN3 mRNA, which leads to the up-regulation of its expression and inhibition of autophagic activity, thus accelerating the development of AP models in mice [68]. This finding not only reveals the pathological role of ZKSCAN3 in AP but also suggests that targeting its expression regulation may provide a new strategy for the treatment of AP. The function of ZKSCAN3 is not limited to digestive diseases but also has a potential regulatory role in respiratory diseases.

Acute lung injury (ALI) is a critical condition triggered by both intrapulmonary and intrapulmonary injurious factors, characterized by lung inflammation, edema, and destruction of alveolar-capillary membranes, ultimately leading to acute hypoxic respiratory failure [122]. Studies have shown that ZKSCAN3 plays a key protective role in lipopolysaccharide (LPS)-induced lung injury. When ZKSCAN3 expression was inhibited, lung tissue injury was significantly aggravated in the LPS mouse model by a mechanism involving decreased antioxidant defenses and increased tissue necrosis [83,123]. It is important to note that ZKSCAN3 is not only involved in the regulation of lung tissue growth and development but also correlates with the level of residual lung function in patients with chronic obstructive pulmonary disease (COPD) [83,124]. The expression of ZKSCAN3 in the lung tissues of COPD patients was negatively correlated with the degree of lung function decline. These findings suggest that the expression level of ZKSCAN3 may become a biomarker for prognostic assessment of ALI, providing new opportunities for the diagnosis and treatment of ALI.

Pathologic association of lung infection with sepsis reveals a central role for ZKSCAN3 in immunomodulation through regulation of the autophagy-lysosome pathway. Studies have shown that in a mouse model of Pseudomonas aeruginosa-induced pneumonia, phagosomal and lysosomal depletion is accompanied by a decrease in Microtubule-associated protein 1A/1B-light chain 3-II (LC3-II) and Lysosome-associated membrane protein 1 (LAMP1) levels, along with enhanced Transcription Factor E3 (TFE3) binding, decreased TFEB, and elevated ZKSCAN3 expression, leading to suppression of autophagy and lysosomal gene transcription [114,125,126]. In contrast, ZKSCAN3 knockout mice exhibit enhanced autophagic lysosomal activity and pathogen clearance in the lungs [127]. This mechanism is not limited to the lungs; the alcohol metabolite acetaldehyde exacerbates HIV-associated hepatocyte injury by up-regulating ZKSCAN3 to inhibit TFEB activity and block lysosomal repair [128]. Notably, secondary death in sepsis survivors is mostly triggered by pulmonary infection and respiratory failure, and abnormalities in autophagy markers (e.g., LC3-II, LAMP1) suggest that lysosomal dysfunction may be a key component of immunosuppression [127]. Together, these findings suggest that ZKSCAN3 is involved in the pathological process of immune imbalance and multi-organ damage after lung infection and sepsis by inhibiting lysosomal production and repair, providing a molecular basis for the development of targeted intervention strategies.

ZKSCAN3 plays a key role in a wide range of pathological processes, from neurodegenerative to infectious diseases, by regulating autophagy and inflammatory responses (Table 2). Its potential value as a therapeutic target has been reported, and new therapies targeting ZKSCAN3 are expected to open up new avenues for the treatment of a wide range of diseases as mechanistic studies intensify. Another study found that the function of ZKSCAN3 in human cells is more inclined to cancer-related pathway regulation rather than directly mediating autophagy or lysosomal generation, and its mechanism of action may be closely related to the cell type and the mode of gene intervention [1]. ZKSCAN3 also has a regulatory role in tumor cells.

## 6. ZKSCAN3 Bridges Cancer, Neurodegenerative Disorders, and Metabolic Diseases Through Autophagic Regulation

ZKSCAN3, as a key suppressor of autophagy, regulates disease pathogenesis by binding to target gene promoters and inhibiting the autophagy-lysosome pathway. This transcription factor serves as a molecular bridge connecting cancer, neurodegenerative disorders, and metabolic diseases through its autophagy-modulating functions. In most malignancies, nuclear accumulation of ZKSCAN3 suppresses autophagy to promote tumor proliferation and metastasis [6]. In neurodegenerative conditions such as AD and PD, its nuclear localization exacerbates toxic protein aggregation by impairing autophagic clearance mechanisms, thereby accelerating disease progression [99,110,129]. Within metabolic disorders, including T2DM-AS, high-glucose stress activates ZKSCAN3, leading to its nuclear retention and subsequent inhibition of macrophage autophagy, which accelerates atherosclerotic plaque formation [117]. In sepsis pathogenesis, ZKSCAN3 up-regulation suppresses LC3-II and LAMP1 expression, compromising lysosomal activity and pathogen clearance capacity [127]. These regulatory mechanisms intersect with upstream signaling pathways such as the p38 MAPK/JNK stress-response axis and epigenetic modulators including ALKBH5. Notably, SIRT1-mediated deacetylation dynamically regulates ZKSCAN3 nuclear localization, establishing a cross-disease regulatory network [117,119]. The multifaceted involvement of ZKSCAN3 across disease spectrums highlights its therapeutic potential as a pharmacological target, with strategies focusing on either direct ZKSCAN3 modulation or intervention in its regulatory partners, providing a conceptual framework for developing pan-disease therapeutic approaches.

## 7. Conclusion and Prospects

This article systematically reviews the biological functions of ZKSCAN3 and its regulatory mechanisms. ZKSCAN3 is involved in the regulation of normal physiological processes such as erythropoiesis, stem cell senescence, and autophagy in cardiomyocytes and retinal epithelial cells. ZKSCAN has also been shown to play a potential role in a variety of diseases, such as PD, SBMA, and AP. In tumors, ZKSCAN3 drives malignant progression by promoting proliferation, invasion, and migration; inhibiting apoptosis and regulating autophagy; and is associated with the activation of multiple oncogenic signaling pathways [5]. Despite significant progress in elucidating its structural features, regulatory networks, and multifunctionality, current research faces several key limitations that require future resolution.

Unclear mechanisms underlying functional duality and context dependency. ZKSCAN3 demonstrates contradictory or sometimes contrasting functions in different normal and disease-related situations [130]. For instance, it inhibits autophagy in most tumors yet activates it in pancreatic cancer [62]. The molecular basis of this high degree of context dependency remains incompletely understood. The specific structural changes, co-regulator interactions, and signal integration mechanisms governing its functional switch (e.g., from an autophagy repressor to a potential activator) are still obscure. Recent evidence suggests that ZKSCAN3 does not play a major part in controlling autophagy gene expression within living organisms [130]. This observation contradicts previous in vitro studies and may be attributed to various factors, including species differences, model discrepancies, and variations in physiological conditions [130]. Future studies should employ multi-omics approaches—such as spatial transcriptomics, single-cell sequencing, and interactomics—combined with cell type-specific gene editing models to precisely delineate ZKSCAN3’s functional network within specific tissue microenvironments or pathological states. Utilizing structural biology techniques, such as cryo-electron microscopy (cryo-EM), to resolve conformational changes in ZKSCAN3 and its complexes under different activation or inhibition states is also crucial.

Insufficient depth in understanding structure–function relationships. Although the domains of ZKSCAN3 (SCAN, KRAB, zinc fingers) have been identified, high-resolution mechanistic studies are lacking regarding how post-translational modifications (PTMs; e.g., phosphorylation, ubiquitination, acetylation) dynamically regulate its subcellular localization, DNA-binding specificity, and interactions with chromatin remodeling complexes (e.g., KAP1/SETDB1/HP1) or transcriptional co-activators/co-repressors [14,104]. Future research could apply advanced biophysical techniques—such as hydrogen–deuterium exchange mass spectrometry (HDX-MS), cross-linking mass spectrometry (XL-MS), and cryo-EM—combined with site-directed mutagenesis to comprehensively map PTM sites and their functional consequences. Developing molecular probes that mimic specific modification states or conformations will further aid in understanding its dynamic regulation.

Discrepancy between animal models and human cell studies. Notably, systemic ZKSCAN3 knockout (KO) mouse models generated using CRISPR/Cas9 technology display relatively normal phenotypes, with no significant enhancement in autophagy activity or target gene expression [130,131]. This contrasts sharply with the potent inhibitory effects of ZKSCAN3 on autophagy and lysosomal gene expression observed in in vitro cell studies. This discrepancy may stem from species differences (only 80% homology between human and mouse ZKSCAN3 protein), compensatory mechanisms following gene knockout, or differences between acute gene silencing (e.g., siRNA/shRNA) and stable knockout effects. Future work could develop tissue- or cell type-specific, inducible ZKSCAN3 KO/knock-in mouse models to more accurately mimic its roles in specific pathophysiological contexts. Concurrently, increased utilization of human organoids, primary cells, or patient-derived xenograft models is needed to bridge the gap between animal models and human disease research.

Challenges and opportunities in targeted therapy. Although ZKSCAN3 is proposed as a potential therapeutic target for various diseases, particularly cancer, direct small-molecule inhibitors targeting ZKSCAN3 are currently unavailable. Existing strategies—such as SIRT1 activators promoting its nuclear export, miR-124 mimics degrading its mRNA, and plant-derived compounds HEP14/HEP15 inhibiting its activity via PKCδ—are indirect and carry risks of off-target effects [104,118,119]. Its functional duality further complicates therapeutic targeting. Future solutions could focus on high-throughput drug screening or structure-based rational drug design targeting key functional domains of ZKSCAN3 (e.g., its DNA-binding domain, KRAB repression domain, or nuclear localization signal) to develop specific small-molecule inhibitors or PROTAC degraders. Simultaneously, in-depth investigation into its mechanisms of action across different disease contexts will help identify safer and more effective therapeutic windows and strategies, such as combination therapies.

Complexity of epigenetic and non-coding RNA regulatory networks. The regulation of ZKSCAN3 levels involves precise control through epigenetic mechanisms such as promoter methylation, histone changes like H3K9me3, non-coding RNAs including miR-124, and epitranscriptomic modifications such as m^6^A [68]. However, how these regulatory layers coordinate within specific cell types or disease states and how ZKSCAN3 functions as an epigenetic regulator influencing global gene expression remain to be fully elucidated. Furthermore, its role in normal human stem cells—such as its regulation of erythrocyte homeostasis via *Tiam1* transcription—and its potential synergistic mechanisms with transcription factors like TFE3 and TFEB need clarification [18,35,127,132]. Future research could employ integrated epigenomic approaches (ChIP-seq, ATAC-seq, MeDIP-seq), epitranscriptomics (m^6^A-seq), and functional genomics screening (CRISPRi/a) to comprehensively map the upstream and downstream regulatory networks of ZKSCAN3 and their dysregulation in disease.

In conclusion, overcoming these limitations will require interdisciplinary collaboration, integrating techniques from structural biology, chemical biology, computational modeling, and advanced disease models. A deeper understanding of the molecular basis of ZKSCAN3’s context-dependent functions and the development of precise targeting tools will be critical for realizing its potential as a diagnostic biomarker and therapeutic target.

## Figures and Tables

**Figure 1 biomolecules-15-01016-f001:**
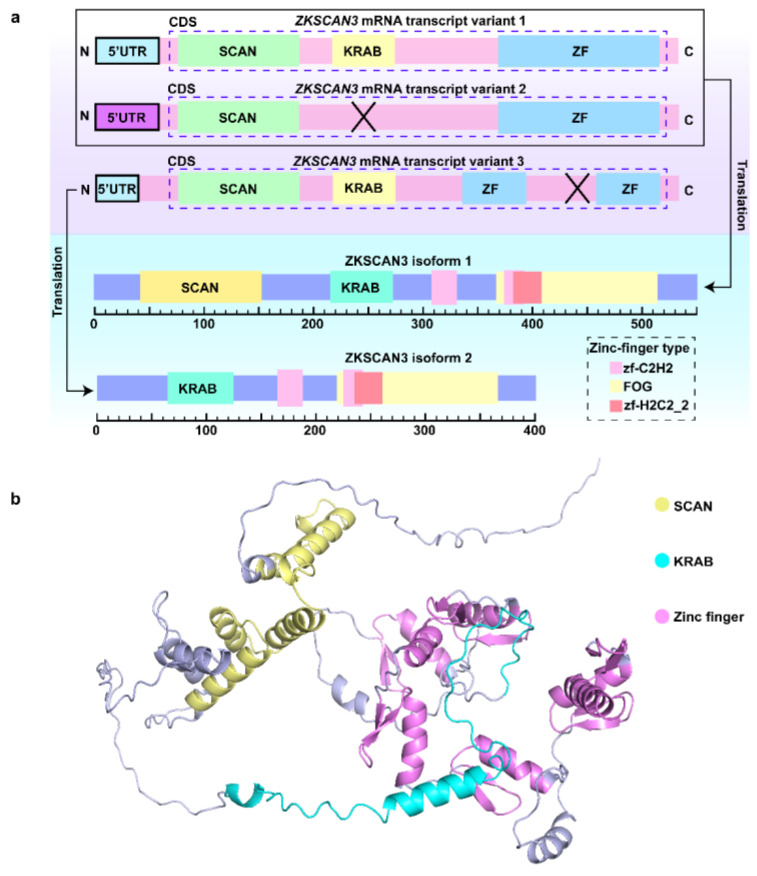
Schematic diagram of *ZKSCAN3* mRNA transcript variants and corresponding ZKSCAN3 isoforms. (**a**) *ZKSCAN3* mRNA transcript variants 1 and 2 are translated into ZKSCAN3 isoform 1, and *ZKSCAN3* mRNA transcript variant 3 is translated into ZKSCAN3 isoform 2. Its protein isoforms have either SCAN structural domains or KRAB structural domains, with different types of zinc-finger structures. (**b**) Structural overview of ZKSCAN3. The structure was predicted by AlphaFold (Accession: Q9BRR0). The SCAN domain (residues 46–128) is highlighted in red; the KRAB domain (residues 214–274) is shown in yellow; and zinc finger domains are depicted in blue, corresponding to residues 314–336, 342–364, 370–392, 398–420, 426–448, 480–502, and 508–530.

**Figure 2 biomolecules-15-01016-f002:**
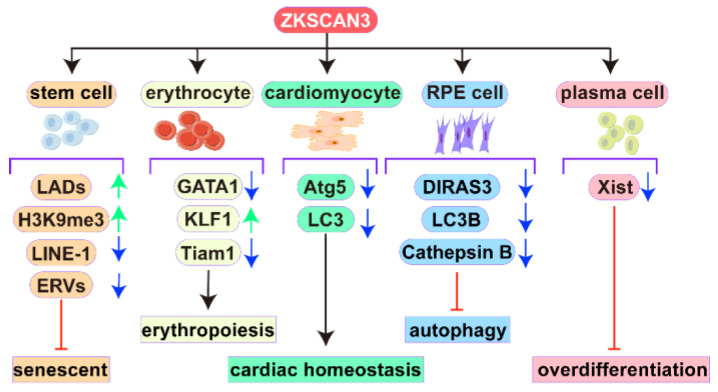
Regulatory roles of ZKSCAN3 in diverse normal cells. In stem cells, ZKSCAN3 stabilizes LADs, promotes the repressive histone modification H3K9me3, suppresses the activation of transposable elements (LINE-1 and ERVs), and ultimately delays cellular senescence. In erythroid cells, ZKSCAN3 inhibits the transcriptional activity of GATA1, activates KLF1 expression, and regulates Tiam1 gene expression to modulate erythropoiesis. In cardiomyocytes, ZKSCAN3 represses autophagy-related genes (*LC3* and *Atg5*), thereby inhibiting autophagy and maintaining cardiac homeostasis. In RPE cells, ZKSCAN3 dynamically regulates autophagy: under nutrient-rich conditions, it localizes to the nucleus and suppresses autophagy-associated genes (*LC3B*, *MAP1LC3B*, and *Cathepsin B*); during stress, it translocates to the cytoplasm to activate autophagic flux. In plasma cells, ZKSCAN3 suppresses Xist expression, thereby limiting excessive plasma cell differentiation.

**Figure 3 biomolecules-15-01016-f003:**
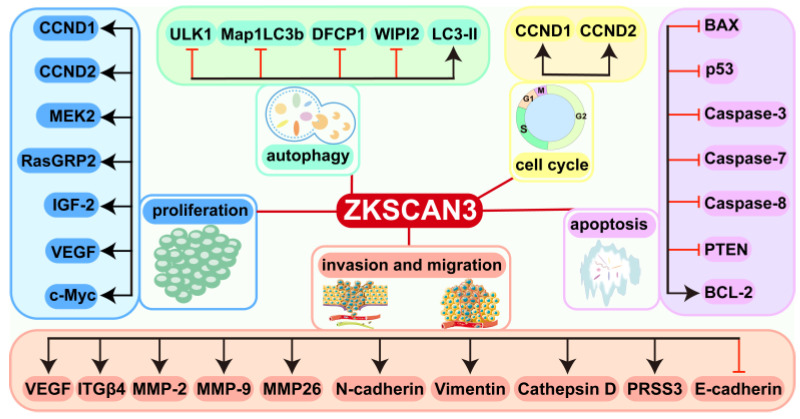
Classification summary of ZKSCAN3-related regulators in cancer. In most tumors ZKSCAN3 plays a pro-carcinogenic role, and at its high expression, it is anti-apoptotic, inhibits autophagy, enhances invasion and migration, and promotes proliferation, cell cycle, and other processes. Special attention should be paid to the fact that in pancreatic cancer, ZKSCAN3 contributes to the up-regulation of LC3-II, which activates autophagy and thus inhibits tumor growth.

**Figure 4 biomolecules-15-01016-f004:**
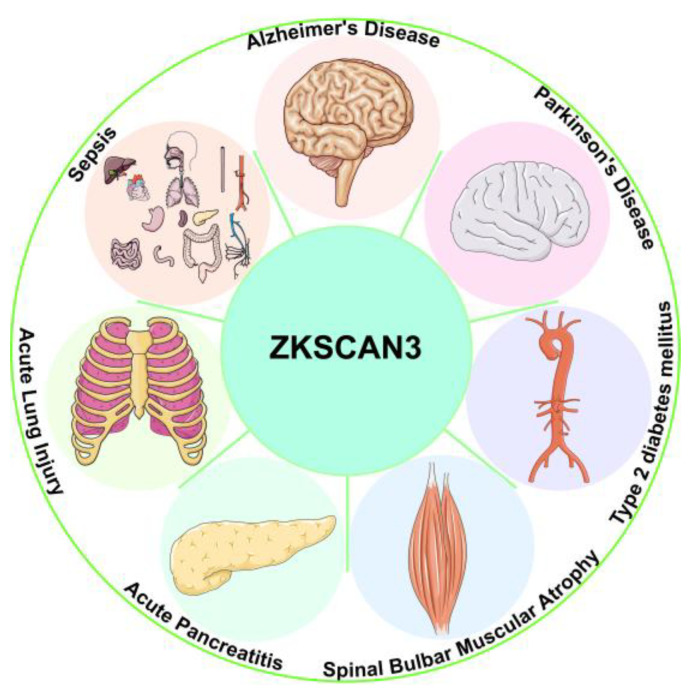
ZKSCAN3 influences the progression of multiple diseases.

**Table 1 biomolecules-15-01016-t001:** Functional regulatory mechanisms and phenotypic roles of ZKSCAN3 in various cancers.

Tumor	Cell Lines	Phenotypes	Mechanisms	Refs
CRC	HCT116, LoVo, LS174T, SW480, RKO, CT26, MCA38	ZKSCAN3 promotes proliferation, invasion, and metastasis	ZKSCAN3 enhances anchorage-independent growth and orthotopic tumor growth; regulates VEGF, ITGβ4, CEA, and AKT expression.	[52,54,57,84]
HCC	Huh7, QGY-7703	ZKSCAN3 inhibits autophagy and promotes migration	ZKSCAN3 binds to the ITGβ4 promoter to activate the FAK/AKT axis; suppresses ULK1 and *LC3b* expression.	[58]
GC	BGC823, HGC-27	ZKSCAN3 promotes proliferation, invasion, metastasis, and tumor angiogenesis	ZKSCAN3 up-regulates MEK2, RasGRP2, IGF-2, and ITGβ4; activates MST1R to induce MMP-2/9, cathepsin D, and VEGF expression.	[69,71]
BCA	MCF-7, MDA-MB-231	ZKSCAN3 regulates cell viability, migration, and invasion	ZKSCAN3 modulates CCND1, BCL-2, MMP-2/9, Bax, and Akt/mTOR signaling pathways.	[63,72,75]
CC	HeLa, C33a, Caski, HeLa, SiHa	ZKSCAN3 promotes cell proliferation	ZKSCAN3 fails to activate autophagy and lysosomal gene expression.	[1,76]
PCa	PC3, LNCaP, VcaP, PC3, DU145, C4-s2	ZKSCAN3 promotes tumor proliferation, angiogenesis, and metastasis	ZKSCAN3 activates VEGF, ITGβ4, cyclin D1/D2, NF-κB, and MMPs.	[74,78]
PaCa	MIA PaCa-2	ZKSCAN3 suppresses cancer cell proliferation, migration, and invasion	ZKSCAN3 activates ULK1 and LC3-II to promote autophagosome formation and lysosomal degradation.	[62,85]
BLCA	UC13, UMUC3, 647V, 5637	ZKSCAN3 regulates autophagy, proliferation, migration, and invasion	Silencing ZKSCAN3 promotes vacuolization, inhibits cell growth, and induces senescence/autophagy Via MMP-2, MMP-9, c-myc/FGFR3, and p53/PTEN.	[6,35,87]

Abbreviations: Colorectal Cancer (CRC), Hepatocellular Carcinoma (HCC), Gastric Cancer (GC), Breast Carcinoma (BCA), Cervical Cancer (CC), Prostate Cancer (PCa), Pancreatic Carcinoma (PaCa), Bladder Carcinoma (BLCA), Multiple Myeloma (MM).

**Table 2 biomolecules-15-01016-t002:** Pathological roles and molecular mechanisms of ZKSCAN3 in non-oncological diseases.

Disease	Mechanisms	Phenotypes	Refs
AD	Down-regulation of ZKSCAN3 leads to up-regulation of GRB2 and Lamp2. Combined intervention of CHIT1 and CGRP inhibits ZKSCAN3.	Enhanced lysosomal activity, reduced neuronal damage, improved synaptic structure. Reduces Aβ toxicity-induced neuronal apoptosis.	[99,100,101,103]
PD	SIRT1 deacetylates ZKSCAN3, thereby promoting its nuclear export. ZKSCAN3 suppresses TFEB nuclear translocation and inhibits *LAMP1* expression. *LRRK2* mutations activate TFEB independently of the mTOR-ZKSCAN3 pathway.	Restored autophagy flux without alleviating α-synuclein aggregation. Targeting SIRT1-ZKSCAN3 axis or TFEB activation may mitigate neuronal damage.	[103,104,106,110,115,116]
T2DM-AS	CEL regulates ZKSCAN3 acetylation through the RAGE/LKB1/AMPK1/SIRT1 pathway.	Impaired macrophage autophagy and reduced plaque stability.	[117]
SBMA	HEP14/HEP15 induces TFEB nuclear translocation and suppresses ZKSCAN3 activity Via JNK/p38 MAPK signaling.	Enhanced lysosomal biogenesis and improved autophagic function.	[15,118,119]
AP	ALKBH5-mediated ZKSCAN3 mRNA demethylation causes its up-regulation.	Reduced autophagic activity and accelerated AP progression.	[68]
ALI	ZKSCAN3 inhibition leads to decreased antioxidant defense and exacerbated tissue necrosis. Negative correlation between ZKSCAN3 expression and lung function in COPD patients.	Aggravated LPS-induced lung injury.	[123]
Sepsis	ZKSCAN3 up-regulation suppresses LC3-II and Lamp1 expression. Acetaldehyde blocks lysosomal repair Via ZKSCAN3.	Reduced autophagic-lysosomal activity and impaired pathogen clearance. Immune dysregulation and multi-organ damage.	[125,127,128]

Abbreviations: Alzheimer’s Disease (AD), Parkinson’s Disease (PD), Type 2 Diabetes Mellitus-Related Atherosclerosis (T2DM-AS), Spinal and Bulbar Muscular Atrophy (SBMA), Acute Pancreatitis (AP), Acute Lung Injury (ALI).

## Data Availability

This is a review article, and no new datasets were generated or analyzed. All data discussed in this manuscript are available in the cited references.

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
