# Peer review of "Research Progress on the Functional Regulation Mechanisms of ZKSCAN3"

_biomolecules, 2025, doi:10.3390/biom15071016_

Round 1

Reviewer 1 Report

Comments and Suggestions for Authors

The review aims to overview recent advances in studies of ZKSCAN3, which is a member of KRAB-domain containing Zn-finger protein family. ZKSCAN3 proteins are composed of SCAN-domain at N-terminus, KRAB-domain and an array of Zn-finger units in C-terminal part.

These transcription factors are involved in regulation of many cellular processes, intensively studied is their role in autophagy and cell cycle progression, therefore proteins play important role in tumorgenesis. Recent results on the role of ZKSCAN3 in physiological and oncogenic processes are the main topic of this review. As to the parts 3-5 of this review, there is a plenty of experimental results presented. Especially useful are Tables. Figures are also informative, especially for the readers new in the field.

Some typos I have found, such as lack of the capital letter at the beginning of sentence (Lines 131, 285, 663). As a not specialist in this field, I was only confused with several references in the Introduction and Part 2:

Line 52-54

"This modular architecture allows ZKSCAN3 to dynamically engage chromatin-remodeling complexes and epigenetic modifiers, thereby influencing gene expression profiles in a context-dependent manner [7]."

The ref. 7 is somewhat strange in this context, at least from the first glance, since they claim in the first lines that "...In vivo, we do not find evidence that this factor is an important regulator of autophagic or lysosomal gene expression...". Maybe it would be better to cite some review here, and ref. 7 could be mentioned in the parts below.

Line 54-57

"... Recent crystallographic and biochemical studies..."

In refs [11, 14, 15] I failed to find any info on crystallographic studies. Simplest search in the PDB revealed 3 PDB entries for SCAN-domains (PDB ID: 3LHR, domain from ZNF24; PDB ID: 4E6S, crystal structure of SCAN-domain from mouse ZFP206 and PDB ID: 2FI2, NMR structure of ZNF174 domain). As to the KRAB-domains, there is at least 2 structures (PDB ID: 8J8N, cryo-em structure of KRAB-domain of ZNF568 with p53 DBD and PDB ID: 7Z36, crystal structure of KAP1 tripartite motif in complex with the ZNF93 KRAB domain). Albeit, these structures are not directly connected with ZKSCAN3 group, but they illustrate structural features of both domains, demonstrated potential of SCAN-domains to dimerise and ability to interact with other proteins, and revealed the interaction of KRAB-domain with KAP1, that allowed to propose a model of silencing complex. Therefore, I would suggest to either remove any mention of structural and crystallographic data from the text, as it is out of the topic, or add structural information in the Part 2, where it would add to the description of SCAN and KRAB-domains which is actually, not very convincing. Also the title could be changed accordingly, since words "structural characteristics" are misleading in this paper.

Line 92-94

"The ZKSCAN3 is localized on chromosome 6 (6p22. 1) and encodes a member of the C2H2-type zinc-finger transcription factor family containing KRAB and SCAN structural domains [6, 21-24].

The domain structure and chromosomal location of protein is not mentioned in refs. 23, and really only ref. 22, which is a review, is OK, I think.

Line 94-97

"Among its protein function modules, the KRAB structural domain plays a pro- or oncogenic role in biological processes such as cell proliferation, apoptosis, migration, and invasion through epigenetic regulation, transcriptional repression or activation, and interactions with other proteins [8-10]"

KRAB domain is described in a single sentence in a most common terms. Since in following parts there is a lot of information concerning role of KRAB-domain, but not much information about its "structural chracteristics". Authors could describe in more details structural features of this relatively small domain. Something like it is a short ~85 aa alpha-helical domain, partially disordered, etc. Maybe it is worth to mention here its interactions with KAP 1?

Line 110-112

"The ZKSCAN3 produces three mRNA transcript variants by variable splicing. Variant 1 contains the complete SCAN, KRAB and zinc finger structural domains and translationally generates the longer protein isoform 1 [28]"

As reported previously, ZER6 exists as two isoforms: shorter p52-ZER6 and longer p71-ZER6. In [28] the role of shorter variant p52 in G6PD regulation is reported. How ZKSCAN3 is related with ZER6 in [28]? May be it would be useful to describe transcript variants as well as isoforms in more details. Whether p52-ZER6 is described in [28] does not contain SCAN-domain (HUB-1?)? I am not a specialist in the field, but this topic needs some more explanation. I believe, that alternative names exist, but here it could be mentioned for clarity. By the way, in some papers longer isoform p71-ZER6 (SCAN-containing?) is inactive in interaction with TEs.

Line 266-268

Along with Fig. 3 which summarizes the signaling pathways influenced by ZKSCAN3 in carcinogenesis, could it be useful to add similar figure combining data on pathways that activate/inhibit expression of ZKSCAN3? Or just reorganize the text in subtopics: regulation of expression of ZK.. and its function in certain types of tumors?

Line271

What is the meaning of "xxx" (...VEGF (xxx)...)? In the cited abstract [57] there are more signaling agents mentioned.

Line 285

"...DLD1/T84, respectively), whereas that of hepatocytes only decreased to 71. 6%. the..."

Typo.

What is DLD1/T84? Cell lines?

Figure 4

I think that there is enough place to put full names of diseases at edges, figure will look better and legend will be shorter.

Author Response

Response to Reviewer comments (Manuscript ID: biomolecules-3681441)

Title: Research Progress on the Functional Regulation Mechanisms of ZKSCAN3

Reviewer 1:

Jianxiong et al. wrote a comprehensive manuscript reviewing the protein ZKSCAN3, a multi-functional and critical regulator for several cellular processes. The manuscript covers most recent discoveries and offers brief but impressive conclusions about these new discoveries together with future directions. ZKSCAN3 involves in the progression of multiple diseases including cancers also provide another perspective of biomarker for diagnose or therapy development.

In conclusion, the authors provide a detailed summary for understanding the complexity of the protein which implicates the requirement of precision regulation. In general, the manuscript is in good shape and will have good impact for the field. However, there are some minor problems and some suggestions that may help improve the manuscript.

Thank you very much for your time and effort reviewing our paper and your positive comments. We had followed your suggestions to improve the quality of our paper. We have superimposed our answers to your comments and suggestions to facilitate the revision. These are our answers to your inquiries:

  1. The title contains “Structural Characteristics” and the author does provide the “Gene Structure and Transcript Variants” section, but the real protein structure is not shown anywhere. The author should show 3D structure to improve the brief functional domains introduction.

Answer:

Thank you sincerely for this constructive suggestion, which has been invaluable in refining the manuscript. In response to Reviewer 3's suggestion, and given that ZKSCAN3 currently only has an AlphaFold-predicted 3D structure without experimental validation, the term "structural characteristics" has been removed from the title and throughout the manuscript to enhance clarity and rigor. A schematic representation of the ZKSCAN3 structure has been added as shown below (2. Gene Structure and Transcript Variants of ZKSCAN3; Revised manuscript pages 4, lines 137-145)

Figure 1b. Structural overview of ZKSCAN3. The structure was predicted by AlphaFold (Accession: Q9BRR0). The SCAN domain (residues 46–128) is highlighted in red; the KRAB domain (residues 214–274) is shown in yellow; and zinc finger domains are depicted in blue, corresponding to residues 314–336, 342–364, 370–392, 398–420, 426–448, 480–502, and 508–530.

  1. Figure 2 need to be optimized to make more sense: the arrows to cells at bottom is not in logic, eg., senescent is not generating stem cell; Atg5/LC3 will be downregulated with ZKSCAN3 deletion, but the red arrow is going up; Some has arrow, some doesn’t. The figure caption need to explain more.

Answer:

Thank you sincerely for your constructive suggestion. We have revised Figure 2 and provided a more detailed explanation in the figure legend, as shown below (3. ZKSCAN3 Exerts Multifunctional Regulatory Mechanisms in Normal Physiological Processes; Revised manuscript pages 5, lines 154-164)

Figure 2. Regulatory roles of ZKSCAN3 in diverse normal cells. In stem cells, ZKSCAN3 stabilizes LADs, promotes the repressive histone modification H3K9me3, suppresses the activation of transposable elements (LINE-1 and ERVs), and ultimately delays cellular senescence. In erythroid cells, ZKSCAN3 inhibits the transcriptional activity of GATA1, activates KLF1 expression, and regulates Tiam1 gene expression to modulate erythropoiesis. In cardiomyocytes, ZKSCAN3 represses autophagy-related genes (LC3 and Atg5), thereby inhibiting autophagy and maintaining cardiac homeostasis. In RPE cells, ZKSCAN3 dynamically regulates autophagy: Under nutrient-rich conditions, it localizes to the nucleus and suppresses autophagy-associated genes (LC3B, MAP1LC3B, and Cathepsin B); During stress, it translocates to the cytoplasm to activate autophagic flux. In plasma cells, ZKSCAN3 suppresses Xist expression, thereby limiting excessive plasma cell differentiation.

  1. Some unfinished marks, eg., line160, “SASP (xxxxx)”; line271, “VEGF (xxx)”

Answer:

Thank you sincerely for your constructive suggestion. Due to an oversight on our part, we failed to remove the redundant content ‘(xxxxx)’, which has now been revised. We sincerely apologize for this careless error and appreciate your patience in bringing it to our attention.

  1. Some non-uniformed reference, eg., line211 “[45]” is superscript.

Answer:

Thank you sincerely for your constructive suggestion. Due to a mistake on my part, I did not format the citation in a uniform manner, which has now been corrected, and we were really sorry for our careless mistakes. Thank you for your reminder.

  1. All the percentage number need to be formed. Should be no space after dot. Eg., line284 29. 9%

Answer:

Thank you sincerely for your constructive feedback. Due to an oversight on our part, the percentages in the manuscript were initially formatted incorrectly. This has now been corrected, and we sincerely apologize for the error. We value your diligence in highlighting this issue.

  1. More abbreviations needed, eg., LADs, MEK2…

Answer:

Thank you sincerely for your constructive suggestion. As recommended, We have now incorporated all abbreviations into the dedicated table at the end of the manuscript, as detailed in the Abbreviations section (revised manuscript pages 19–20, lines 776–777).

  1. Line265 “specific recognition of DNA motifs” should be “specific DNA recognition motif”.

Answer:

Thank you sincerely for your constructive suggestion. We have revised it in the manuscript to read as follows (4.1 ZKSCAN3 Drives Colorectal Cancer Progression by Activating Wnt/β-Catenin and ITGβ4/FAK/AKT Pathways; Revised manuscript pages 8, lines 283-284) : ZKSCAN3, plays a regulatory role in colorectal carcinogenesis. It regulates the expression of 280 target genes (204 up-regulated and 76 down-regulated) through specific DNA recognition motif (KRDGGG), several of which are closely related to cell proliferation and survival [1, 2].

We greatly appreciate your comments to improve this paper, and hope that these revisions will make our manuscript suitable for publication in Biomolecules.

References

  1. Yang, L.; Zhang, L.; Wu, Q.; Boyd, D.D. Unbiased Screening for Transcriptional Targets of ZKSCAN3 Identifies Integrin β4 and Vascular Endothelial Growth Factor as Downstream Targets. Journal of Biological Chemistry. 2008;283(50):35295-35304.
  2. Qian, Z.; Chang, T.; Zhang, T.; Wang, J.; Gu, H. Genomic analyses identify biological processes in ZKSCAN3-deficient colorectal cancer cells. bioRxiv. 2022:2021.2012. 2030.474589.

Reviewer 2 Report

Comments and Suggestions for Authors

This manuscript describes the structural and functional features of ZKSCAN3 comprehensively. Authors have summarized a variety of literature and organized the content logically. The review article should serve as a valuable reference for the field. However, several issues should be addressed to improve the quality of the review. Below are my comments/questions.
1. This manuscript summarized a lot of information, but needs more critical evaluation. Please add limitations of the research so far studied and how such things can be resolved, if possible.
2. Is it possible to make a section dealing with mechasisms of ZKSCAN3 common throughout cancer, neurodegeneration, and metabolic disease?

Author Response

Response to Reviewer comments (Manuscript ID: biomolecules-3681441)

Title: Research Progress on the Functional Regulation Mechanisms of ZKSCAN3

Reviewer 2:

This manuscript describes the structural and functional features of ZKSCAN3 comprehensively. Authors have summarized a variety of literature and organized the content logically. The review article should serve as a valuable reference for the field.

Thank you very much for your time and effort reviewing our paper and your positive comments. We had followed your suggestions to improve the quality of our paper. We have superimposed our answers to your comments and suggestions to facilitate the revision. These are our answers to your inquiries:

However, several issues should be addressed to improve the quality of the review. Below are my comments/questions.

  1. This manuscript summarized a lot of information, but needs more critical evaluation. Please add limitations of the research so far studied and how such things can be resolved, if possible.

Answer:

Thank you sincerely for your constructive suggestion. The content regarding the limitations of the research conducted so far and possible ways to address them has been added. (7. Conclusion and Prospects; Revised manuscript pages 17-19, lines 681-757):

Despite significant progress in elucidating its structural features, regulatory networks, and multifunctionality, current research faces several key limitations that require future resolution.

Unclear mechanisms underlying functional duality and context dependency. ZKSCAN3 demonstrates contradictory or sometimes contrasting functions in different normal and disease-related situations [1]. For instance, it inhibits autophagy in most tumors yet activates it in pancreatic cancer [2]. The molecular basis of this high degree of context dependency remains incompletely understood. The specific structural changes, co-regulator interactions, and signal integration mechanisms governing its functional switch (e.g., from an autophagy repressor to a potential activator) are still obscure. Recent evidence suggests that ZKSCAN3 does not play a major part in controlling autophagy gene expression within living organisms [1]. This observation contradicts previous in vitro studies and may be attributed to various factors, including species differences, model discrepancies, and variations in physiological conditions [1]. Future studies should employ multi-omics approaches—such as spatial transcriptomics, single-cell sequencing, and interactomics—combined with cell type-specific gene editing models to precisely delineate ZKSCAN3’s functional network within specific tissue microenvironments or pathological states. Utilizing structural biology techniques, such as cryo-electron microscopy (cryo-EM), to resolve conformational changes of ZKSCAN3 and its complexes under different activation or inhibition states is also crucial.

Insufficient depth in understanding structure-function relationships. Although the domains of ZKSCAN3 (SCAN, KRAB, zinc fingers) have been identified, high-resolution mechanistic studies are lacking regarding how post-translational modifications (PTMs; e.g., phosphorylation, ubiquitination, acetylation) dynamically regulate its subcellular localization, DNA-binding specificity, and interactions with chromatin remodeling complexes (e.g., KAP1/SETDB1/HP1) or transcriptional co-activators/co-repressors[3, 4]. Future research could apply advanced biophysical techniques—such as hydrogen-deuterium exchange mass spectrometry (HDX-MS), cross-linking mass spectrometry (XL-MS), and cryo-EM—combined with site-directed mutagenesis, to comprehensively map PTM sites and their functional consequences. Developing molecular probes that mimic specific modification states or conformations will further aid in understanding its dynamic regulation.

Discrepancy between animal models and human cell studies. Notably, systemic ZKSCAN3 knockout (KO) mouse models generated using CRISPR/Cas9 technology display relatively normal phenotypes, with no significant enhancement in autophagy activity or target gene expression [1, 5]. This contrasts sharply with the potent inhibitory effects of ZKSCAN3 on autophagy and lysosomal gene expression observed in in vitro cell studies. This discrepancy may stem from species differences (only 80% homology between human and mouse ZKSCAN3 protein), compensatory mechanisms following gene knockout, or differences between acute gene silencing (e.g., siRNA/shRNA) and stable knockout effects. Future work could develop tissue- or cell type-specific, inducible ZKSCAN3 KO/knock-in mouse models to more accurately mimic its roles in specific pathophysiological contexts. Concurrently, increased utilization of human organoids, primary cells, or patient-derived xenograft models is needed to bridge the gap between animal models and human disease research.

Challenges and opportunities in targeted therapy. Although ZKSCAN3 is proposed as a potential therapeutic target for various diseases, particularly cancer, direct small-molecule inhibitors targeting ZKSCAN3 are currently unavailable. Existing strategies—such as SIRT1 activators promoting its nuclear export, miR-124 mimics degrading its mRNA, and plant-derived compounds HEP14/HEP15 inhibiting its activity via PKCδ—are indirect and carry risks of off-target effects [3, 6, 7]. Its functional duality further complicates therapeutic targeting. Future solutions could focus on high-throughput drug screening or structure-based rational drug design targeting key functional domains of ZKSCAN3 (e.g., its DNA-binding domain, KRAB repression domain, or nuclear localization signal) to develop specific small-molecule inhibitors or PROTAC degraders. Simultaneously, in-depth investigation into its mechanisms of action across different disease contexts will help identify safer and more effective therapeutic windows and strategies, such as combination therapies.

Complexity of epigenetic and non-coding RNA regulatory networks. The regulation of ZKSCAN3 levels involves precise control through epigenetic mechanisms such as promoter methylation, histone changes like H3K9me3, non-coding RNAs including miR-124, and epitranscriptomic modifications such as m⁶A [8]. However, how these regulatory layers coordinate within specific cell types or disease states, and how ZKSCAN3 functions as an epigenetic regulator influencing global gene expression, remains to be fully elucidated. Furthermore, its role in normal human stem cells—such as its regulation of erythrocyte homeostasis via Tiam1 transcription—and its potential synergistic mechanisms with transcription factors like TFE3 and TFEB need clarification [9-12]. Future research could employ integrated epigenomic approaches (ChIP-seq, ATAC-seq, MeDIP-seq), epitranscriptomics (m⁶A-seq), and functional genomics screening (CRISPRi/a) to comprehensively map the upstream and downstream regulatory networks of ZKSCAN3 and their dysregulation in disease.

In conclusion, overcoming these limitations will require interdisciplinary collaboration, integrating techniques from structural biology, chemical biology, computational modeling, and advanced disease models. A deeper understanding of the molecular basis of ZKSCAN3’s context-dependent functions and the development of precise targeting tools will be critical for realizing its potential as a diagnostic biomarker and therapeutic target.

  1. Is it possible to make a section dealing with mechasisms of ZKSCAN3 common throughout cancer, neurodegeneration, and metabolic disease?

Answer:

Thank you sincerely for your constructive suggestion. We believe it is very necessary to create a section addressing the mechanisms of ZKSCAN3 that are common across cancer, neurodegeneration, and metabolic disease. Below are the specific details (revised manuscript pages 17, lines 651-672):

  1. ZKSCAN3 Bridges Cancer, Neurodegenerative Disorders, and Metabolic Diseases Through Autophagic Regulation

ZKSCAN3, as a key suppressor of autophagy, regulates disease pathogenesis by binding to target gene promoters and inhibiting the autophagy-lysosome pathway. This transcription factor serves as a molecular bridge connecting cancer, neurodegenerative disorders, and metabolic diseases through its autophagy-modulating functions. In most malignancies, nuclear accumulation of ZKSCAN3 suppresses autophagy to promote tumor proliferation and metastasis [13]. In neurodegenerative conditions such as AD and PD, its nuclear localization exacerbates toxic protein aggregation by impairing autophagic clearance mechanisms, thereby accelerating disease progression [14-16]. Within metabolic disorders including T2DM-AS, high-glucose stress activates ZKSCAN3, leading to its nuclear retention and subsequent inhibition of macrophage autophagy, which accelerates atherosclerotic plaque formation [17]. In Sepsis pathogenesis, ZKSCAN3 upregulation suppresses LC3-II and LAMP1 expression, compromising lysosomal activity and pathogen clearance capacity [11]. These regulatory mechanisms intersect with upstream signaling pathways such as the p38 MAPK/JNK stress-response axis and epigenetic modulators including ALKBH5. Notably, SIRT1-mediated deacetylation dynamically regulates ZKSCAN3 nuclear localization, establishing a cross-disease regulatory network [7, 17]. The multifaceted involvement of ZKSCAN3 across disease spectrums highlights its therapeutic potential as a pharmacological target, with strategies focusing on either direct ZKSCAN3 modulation or intervention in its regulatory partners, providing a conceptual framework for developing pan-disease therapeutic approaches.

We greatly appreciate your comments to improve this paper, and hope that these revisions will make our manuscript suitable for publication in Biomolecules.

References

  1. Pan, H.; Yan, Y.; Liu, C.; Finkel, T. The role of ZKSCAN3 in the transcriptional regulation of autophagy. Autophagy. 2017;13(7):1235-1238.
  2. Nonoyama, K.; Matsuo, Y.; Sugita, S.; Eguchi, Y.; Denda, Y.; Murase, H.; Kato, T.; Imafuji, H.; Saito, K.; Morimoto, M., et al. Expression of ZKSCAN3 protein suppresses proliferation, migration, and invasion of pancreatic cancer through autophagy. Cancer Sci. 2024;115(6):1964-1978.
  3. Wu, X.; Ren, Y.; Wen, Y.; Lu, S.; Li, H.; Yu, H.; Li, W.; Zou, F. Deacetylation of ZKSCAN3 by SIRT1 induces autophagy and protects SN4741 cells against MPP(+)-induced oxidative stress. Free Radic Biol Med. 2022;181:82-97.
  4. Groner, A.C.; Meylan, S.; Ciuffi, A.; Zangger, N.; Ambrosini, G.; Dénervaud, N.; Bucher, P.; Trono, D. KRAB-zinc finger proteins and KAP1 can mediate long-range transcriptional repression through heterochromatin spreading. PLoS Genet. 2010;6(3):e1000869.
  5. Liu, Z.; Li, X.; Li, X.; Li, Z.; Chen, H.; Gong, S.; Zhang, M.; Zhang, Y.; Li, Z.; Yang, L., et al. The kidney-expressed transcription factor ZKSCAN3 is dispensable for autophagy transcriptional regulation and AKI progression in mouse. Mutat Res. 2022;825:111790.
  6. Yin, L.; Zhou, J.; Li, T.; Wang, X.; Xue, W.; Zhang, J.; Lin, L.; Wang, N.; Kang, X.; Zhou, Y., et al. Inhibition of the dopamine transporter promotes lysosome biogenesis and ameliorates Alzheimer's disease-like symptoms in mice. Alzheimers Dement. 2023;19(4):1343-1357.
  7. Li, Y.; Xu, M.; Ding, X.; Yan, C.; Song, Z.; Chen, L.; Huang, X.; Wang, X.; Jian, Y.; Tang, G., et al. Protein kinase C controls lysosome biogenesis independently of mTORC1. Nat Cell Biol. 2016;18(10):1065-1077.
  8. Zhang, T.; Zhu, S.; Huang, G.W. ALKBH5 suppresses autophagic flux via N6-methyladenosine demethylation of ZKSCAN3 mRNA in acute pancreatitis. World J Gastroenterol. 2024;30(12):1764-1776.
  9. Li, Z.; Sheng, B.; Zhang, T.; Wang, T.; Chen, D.; An, G.; Wang, X.; Meng, H.; Yang, L. Zkscan3 affects erythroblast development by regulating the transcriptional activity of GATA1 and KLF1 in mice. J Mol Histol. 2022;53(2):423-436.
  10. Yu, B.; Qi, Y.; Li, R.; Shi, Q.; Satpathy, A.T.; Chang, H.Y. B cell-specific XIST complex enforces X-inactivation and restrains atypical B cells. Cell. 2021;184(7):1790-1803.e1717.
  11. Ouyang, X.; Becker, E., Jr.; Bone, N.B.; Johnson, M.S.; Craver, J.; Zong, W.X.; Darley-Usmar, V.M.; Zmijewski, J.W.; Zhang, J. ZKSCAN3 in severe bacterial lung infection and sepsis-induced immunosuppression. Lab Invest. 2021;101(11):1467-1474.
  12. Li, S.; Song, Y.; Quach, C.; Guo, H.; Jang, G.B.; Maazi, H.; Zhao, S.; Sands, N.A.; Liu, Q.; In, G.K., et al. Transcriptional regulation of autophagy-lysosomal function in BRAF-driven melanoma progression and chemoresistance. Nat Commun. 2019;10(1):1693.
  13. Chauhan, S.; Goodwin, J.G.; Chauhan, S.; Manyam, G.; Wang, J.; Kamat, A.M.; Boyd, D.D. ZKSCAN3 is a master transcriptional repressor of autophagy. Mol Cell. 2013;50(1):16-28.
  14. Yang, W.; Yu, W.; Lv, Y. Neuroprotective effects of chitinase-1 and calcitonin gene-related peptide on Alzheimer's disease by promoting lysosomal function. J Alzheimers Dis. 2025;103(3):879-888.
  15. Yang, M.; Lin, S.; Sun, B.; Chen, W.; Liu, J.; Chen, M. ZKSCAN3 affects the autophagy‑lysosome pathway through TFEB in Parkinson's disease. Biomed Rep. 2025;22(4):74.
  16. Levine, B.; Kroemer, G. Autophagy in the pathogenesis of disease. Cell. 2008;132(1):27-42.
  17. Liu, Z.; Liu, J.; Wang, X.; Zhang, Y.; Ma, Y.; Guan, G.; Yuwen, Y.; He, N.; Liu, H.; Yu, X., et al. N(ε)-carboxyethyl-lysin influences atherosclerotic plaque stability through ZKSCAN3 acetylation-regulated macrophage autophagy via the RAGE/LKB1/AMPK1/SIRT1 pathway. Cardiovasc Diabetol. 2025;24(1):36.

Reviewer 3 Report

Comments and Suggestions for Authors

Jianxiong et al. wrote a comprehensive manuscript reviewing the protein ZKSCAN3, a multi-functional and critical regulator for several cellular processes. The manuscript covers most recent discoveries and offers brief but impressive conclusions about these new discoveries together with future directions. ZKSCAN3 involves in the progression of multiple diseases including cancers also provide another perspective of biomarker for diagnose or therapy development.

In conclusion, the authors provide a detailed summary for understanding the complexity of the protein which implicates the requirement of precision regulation. In general, the manuscript is in good shape and will have good impact for the field. However, there are some minor problems and some suggestions that may help improve the manuscript.

  1. The title contains “Structural Characteristics” and the author does provide the “Gene Structure and Transcript Variants” section, but the real protein structure is not shown anywhere. The author should show 3D structure to improve the brief functional domains introduction.
  2. Figure 2 need to be optimized to make more sense: the arrows to cells at bottom is not in logic, eg., senescent is not generating stem cell; Atg5/LC3 will be downregulated with ZKSCAN3 deletion, but the red arrow is going up; Some has arrow, some doesn’t. The figure caption need to explain more.
  3. Some unfinished marks, eg., line160, “SASP (xxxxx)”; line271, “VEGF (xxx)”
  4. Some non-uniformed reference, eg., line211 “[45]” is superscript.
  5. All the percentage number need to be formed. Should be no space after dot. Eg., line284 29. 9%
  6. More abbreviations needed, eg., LADs, MEK2…
  7. Line265 “specific recognition of DNA motifs” should be “specific DNA recognition motif”.

Author Response

Response to Reviewer comments (Manuscript ID: biomolecules-3681441)

Title: Research Progress on the Functional Regulation Mechanisms of ZKSCAN3

Reviewer 3:

The review aims to overview recent advances in studies of ZKSCAN3, which is a member of KRAB-domain containing Zn-finger protein family. ZKSCAN3 proteins are composed of SCAN-domain at N-terminus, KRAB-domain and an array of Zn-finger units in C-terminal part.

These transcription factors are involved in regulation of many cellular processes, intensively studied is their role in autophagy and cell cycle progression, therefore proteins play important role in tumorgenesis. Recent results on the role of ZKSCAN3 in physiological and oncogenic processes are the main topic of this review. As to the parts 3-5 of this review, there is a plenty of experimental results presented. Especially useful are Tables. Figures are also informative, especially for the readers new in the field.

Thank you very much for your time and effort reviewing our paper and your positive comments. We had followed your suggestions to improve the quality of our paper. We have superimposed our answers to your comments and suggestions to facilitate the revision. These are our answers to your inquiries:

  1. Some typos I have found, such as lack of the capital letter at the beginning of sentence (Lines 131, 285, 663).

Answer:

Thanks for your careful checks. We are sorry for our carelessness. Based on your comments, we have made the corrections to make the word harmonized within the whole manuscript.

  1. As a not specialist in this field, I was only confused with several references in the Introduction and Part 2:

Line 52-54

"This modular architecture allows ZKSCAN3 to dynamically engage chromatin-remodeling complexes and epigenetic modifiers, thereby influencing gene expression profiles in a context-dependent manner [7]."

The ref. 7 is somewhat strange in this context, at least from the first glance, since they claim in the first lines that "...In vivo, we do not find evidence that this factor is an important regulator of autophagic or lysosomal gene expression...". Maybe it would be better to cite some review here, and ref. 7 could be mentioned in the parts below.

Answer:

Thank you sincerely for your constructive suggestion. As recommended, we have revised the references to: ‘The Biological Roles of ZKSCAN3 (ZNF306) in the Hallmarks of Cancer: From Mechanisms to Therapeutics.’, ‘ZKSCAN3 is a master transcriptional repressor of autophagy.’. The specific content is as follows (1. Introductionrevised manuscript pages 2, lines 48-50): This modular architecture allows ZKSCAN3 to dynamically engage chromatin-remodeling complexes and epigenetic modifiers, thereby influencing gene expression profiles in a context-dependent manner [1, 2].

Additionally, reference ‘[7]’(‘The role of ZKSCAN3 in the transcriptional regulation of autophagy. Autophagy.’) is mentioned in the Discussion section (7. Conclusion and Prospects; revised manuscript pages 17, lines 690-694): Recent evidence suggests that ZKSCAN3 does not play a major part in controlling autophagy gene expression within living organisms. This observation contradicts previous in vitro studies and may be attributed to various factors, including species differences, model discrepancies, and variations in physiological conditions [3].

  1. Line 54-57

"... Recent crystallographic and biochemical studies..."

In refs [11, 14, 15] I failed to find any info on crystallographic studies. Simplest search in the PDB revealed 3 PDB entries for SCAN-domains (PDB ID: 3LHR, domain from ZNF24; PDB ID: 4E6S, crystal structure of SCAN-domain from mouse ZFP206 and PDB ID: 2FI2, NMR structure of ZNF174 domain). As to the KRAB-domains, there is at least 2 structures (PDB ID: 8J8N, cryo-em structure of KRAB-domain of ZNF568 with p53 DBD and PDB ID: 7Z36, crystal structure of KAP1 tripartite motif in complex with the ZNF93 KRAB domain). Albeit, these structures are not directly connected with ZKSCAN3 group, but they illustrate structural features of both domains, demonstrated potential of SCAN-domains to dimerise and ability to interact with other proteins, and revealed the interaction of KRAB-domain with KAP1, that allowed to propose a model of silencing complex. Therefore, I would suggest to either remove any mention of structural and crystallographic data from the text, as it is out of the topic, or add structural information in the Part 2, where it would add to the description of SCAN and KRAB-domains which is actually, not very convincing. Also the title could be changed accordingly, since words "structural characteristics" are misleading in this paper.

Answer:

Thank you sincerely for your constructive suggestion. As recommended, we have removed all content pertaining to 'structural characteristics' and 'crystallography' throughout the manuscript. The title has been revised to 'Research Progress on the Functional Regulation Mechanisms of ZKSCAN3'. Additionally, the phrase 'Recent crystallographic and biochemical studies...' in the Introduction section (revised manuscript page 2, lines 5054) was replaced with “Although the direct structure of ZKSCAN3 has not been resolved, homologous protein studies of the SCAN/KRAB structural domains suggest that such domains can mediate protein interactions through dimerization or recruit chromatin-modifying complexes (e.g., KRAB-KAP1) to achieve transcriptional repression [4-6].”

  1. Line 92-94

"The ZKSCAN3 is localized on chromosome 6 (6p22. 1) and encodes a member of the C2H2-type zinc-finger transcription factor family containing KRAB and SCAN structural domains [6, 21-24].

The domain structure and chromosomal location of protein is not mentioned in refs. 23, and really only ref. 22, which is a review, is OK, I think.

Answer:

Thank you sincerely for your constructive suggestion. As recommended, we have streamlined the reference list and updated the citations in Section 2 (Gene Structure and Transcript Variants of ZKSCAN3; Revised manuscript page 2, lines 87-88) to The previously undescribed ZKSCAN3 (ZNF306) is a novel “driver” of colorectal cancer progression’, KRAB-containing zinc-finger repressor proteins. The revised text reads as follows (2. Gene Structure and Transcript Variants of ZKSCAN3revised manuscript pages 2, lines 86-87): The ZKSCAN3 is localized on chromosome 6 (6p22. 1) and encodes a member of the C2H2-type zinc-finger transcription factor family containing KRAB and SCAN structural domains [7, 8].

  1. Line 94-97

"Among its protein function modules, the KRAB structural domain plays a pro- or oncogenic role in biological processes such as cell proliferation, apoptosis, migration, and invasion through epigenetic regulation, transcriptional repression or activation, and interactions with other proteins [8-10]"

KRAB domain is described in a single sentence in a most common terms. Since in following parts there is a lot of information concerning role of KRAB-domain, but not much information about its "structural chracteristics". Authors could describe in more details structural features of this relatively small domain. Something like it is a short ~85 aa alpha-helical domain, partially disordered, etc. Maybe it is worth to mention here its interactions with KAP 1?

Answer:

Thank you sincerely for your constructive suggestion. As recommended, We have added to the manuscript (2. Gene Structure and Transcript Variants of ZKSCAN3; Revised manuscript pages 2-3, lines 91-106): The KRAB domain is a modular protein segment comprising approximately 75 to 100 amino acids [8, 9]. Its core consists of a highly conserved approximately 45-amino acid A-box and a less conserved approximately 30-amino acid B-box. The A-box forms two stable α-helices whose hydrophobic core maintains structural stability, while surface-exposed acidic residues play key roles in mediating protein-protein interactions. The B-box indirectly supports overall function by enhancing the activity of the A-box [8, 9]. The KRAB domain contains intrinsically disordered regions at its N-terminal and C-terminal ends to confer conformational flexibility, while its core helical topology is highly conserved across mammals, reflecting its critical functional importance. [9, 10]. In KRAB zinc finger proteins (KRAB-ZFPs), the KRAB-A box interacts with the RBCC domain (comprising RING, B-box, and coiled-coil regions) of KAP1 via hydrophobic interactions [5]. Upon binding, KAP1 dimerizes and recruits chromatin remodeling complexes such as HP1, SETDB1, and NuRD, which modify chromatin through H3K9 trimethylation (H3K9me3) and deacetylation to repress target gene transcription. Furthermore, the RBCC domain of KAP1 facilitates the nuclear import of KRAB-ZFPs through a “piggy-back” mechanism, enabling their function in transcriptional regulation [5, 8-11].

  1. Line 110-112

"The ZKSCAN3 produces three mRNA transcript variants by variable splicing. Variant 1 contains the complete SCAN, KRAB and zinc finger structural domains and translationally generates the longer protein isoform 1 [28]"

As reported previously, ZER6 exists as two isoforms: shorter p52-ZER6 and longer p71-ZER6. In [28] the role of shorter variant p52 in G6PD regulation is reported. How ZKSCAN3 is related with ZER6 in [28]? May be it would be useful to describe transcript variants as well as isoforms in more details. Whether p52-ZER6 is described in [28] does not contain SCAN-domain (HUB-1?)? I am not a specialist in the field, but this topic needs some more explanation. I believe, that alternative names exist, but here it could be mentioned for clarity. By the way, in some papers longer isoform p71-ZER6 (SCAN-containing?) is inactive in interaction with TEs.

Answer:

Thank you sincerely for your constructive suggestion. Through comprehensive literature review, we identified ZER6 as a zinc finger protein gene that generates two major isoforms through alternative splicing:

(1) p52-ZER6 (short isoform): Contains a truncated KRAB domain (tKRAB) + 6 C2H2-type zinc finger motifs. Lacks the HUB-1 domain. Binds to p53/MDM2 complex to promote p53 degradation, exhibiting oncogenic properties [12-14].

(2) p71-ZER6 (long isoform): Features an additional 109-amino acid extension at the N-terminus compared to p52-ZER6, forming a complete KRAB domain and HUB-1 domain. Retains HUB-1 domain which blocks p53 interaction, showing no involvement in p53 regulation but potential cancer-promoting effects through alternative pathways [12-14].

Critical distinctions from ZKSCAN3:

These isoforms show minimal functional overlap with ZKSCAN3, representing distinct genetic entities. While ZKSCAN3 contains a SCAN domain and regulates autophagy, ZER6 primarily operates through the tKRAB-p53 axis. Considering these mechanistic disparities, we have intentionally excluded ZER6-related content from the current manuscript.

As recommended, we have updated the references to 'Effect of Lacking ZKSCAN3 on Autophagy, Lysosomal Biogenesis and Senescence' and 'The Biological Roles of ZKSCAN3 (ZNF306) in the Hallmarks of Cancer: From Mechanisms to Therapeutics'. The specific content in the manuscript is as follows (2. Gene Structure and Transcript Variants of ZKSCAN3; revised manuscript pages 3, lines 119-121): The ZKSCAN3 produces three mRNA transcript variants by variable splicing. Variant 1 contains the complete SCAN, KRAB and zinc finger structural domains and translationally generates the longer protein isoform 1 [1, 15].

  1. Line 266-268

Along with Fig. 3 which summarizes the signaling pathways influenced by ZKSCAN3 in carcinogenesis, could it be useful to add similar figure combining data on pathways that activate/inhibit expression of ZKSCAN3? Or just reorganize the text in subtopics: regulation of expression of ZK.. and its function in certain types of tumors?

Answer:

Thank you sincerely for your constructive suggestion. As recommended, we have incorporated a subtitle into the manuscript after careful consideration, as shown below:

4.1 ZKSCAN3 Drives Colorectal Cancer Progression by Activating Wnt/β-Catenin and ITGβ4/FAK/AKT Pathways (revised manuscript pages 8, lines 280-281)

4.2 ZKSCAN3 Promotes Hepatocellular Carcinoma Progression via the FAK/AKT-Autophagy Inhibition Axis (revised manuscript pages 9, lines 315-316)

4.3 ZKSCAN3 Facilitates Gastric Cancer Progression Through Ras/MAPK-MST1R-MMP/VEGF Multi-Axis Signaling (revised manuscript pages 9, lines 340-341)

4.4 ZKSCAN3 Accelerates Breast Cancer Proliferation and Invasion via AKT/mTOR-Cyclin D1-MMP Axis (revised manuscript pages 9, lines 370-371)

4.5 ZKSCAN3 Drives Cervical Cancer Proliferation and Metastasis Through RAS-MAPK/MST1R/VEGF Pathway (revised manuscript pages 10, lines 388-389)

4.6 ZKSCAN3 Promotes Prostate Cancer Proliferation and Metastasis via VEGF/ITGβ4-Cyclin D-NFκB-MMP Axis (revised manuscript pages 11, lines 411-412)

4.7 ZKSCAN3 Inhibits Malignant Progression of Pancreatic Cancer by Targeting ULK1/LC3-II Autophagy Axis (revised manuscript pages 11, lines 432-433)

4.8 ZKSCAN3 Activates Bladder Cancer Invasion and Proliferation via c-Myc/FGFR3-MMP2/9 Signaling Axis (revised manuscript pages 12, lines 454-455)

4.9 ZKSCAN3 Synergizes with EGFR to Activate Pro-Survival, Anti-Apoptotic, and Tumor Microenvironment Remodeling Pathways in Ovarian Cancer (revised manuscript pages 12, lines 473-474)

4.10 ZKSCAN3 Enhances Cell Cycle Progression and Angiogenesis in Multiple Myeloma by Regulating CCND2 and VEGF (revised manuscript pages 12, lines 495-496)

  1. Line271

What is the meaning of "xxx" (...VEGF (xxx)...)? In the cited abstract [57] there are more signaling agents mentioned.

Answer:

Thank you sincerely for your constructive suggestion. Due to an oversight on our part, we failed to remove the redundant content ‘(xxx)’, which has now been revised. We sincerely apologize for this careless error and appreciate your patience in bringing it to our attention.

  1. Line 285

"...DLD1/T84, respectively), whereas that of hepatocytes only decreased to 71. 6%. the..."

Typo.

What is DLD1/T84? Cell lines?

Answer:

Thank you sincerely for your constructive suggestion. We confirmed that both DLD1 and T84 are colorectal cancer cell lines. As recommended, we have now supplemented and refined the relevant content in the manuscript, as detailed below (4.1 ZKSCAN3 Drives Colorectal Cancer Progression by Activating Wnt/β-Catenin and ITGβ4/FAK/AKT Pathways; Revised manuscript pages 8, lines 301-304Mutation of the ZKSCAN3 binding site in the promoter caused the activity of colon cancer cells to plummet (to 29.9% and 14.3% in DLD1 and T84 cell lines, respectively), whereas the activity of hepatocytes only decreased to 71.6% [16].

  1. Figure 4

I think that there is enough place to put full names of diseases at edges, figure will look better and legend will be shorter.

Answer:

Thank you sincerely for your constructive suggestion. As recommended, We have made changes to the diagram, which are as follows (5. The Pathological Regulatory Role and Therapeutic Potential of ZKSCAN3 in Multiple Diseasesrevised manuscript pages 14, lines 533-534;

Figure 4. ZKSCAN3 influences the progression of multiple diseases.

We greatly appreciate your comments to improve this paper, and hope that these revisions will make our manuscript suitable for publication in Biomolecules.

References

  1. Li, W.; Zhang, H.; Xu, J.; Maimaitijiang, A.; Su, Z.; Fan, Z.; Li, J. The Biological Roles of ZKSCAN3 (ZNF306) in the Hallmarks of Cancer: From Mechanisms to Therapeutics. Int J Mol Sci. 2024;25(21).
  2. Chauhan, S.; Goodwin, J.G.; Chauhan, S.; Manyam, G.; Wang, J.; Kamat, A.M.; Boyd, D.D. ZKSCAN3 is a master transcriptional repressor of autophagy. Mol Cell. 2013;50(1):16-28.
  3. Pan, H.; Yan, Y.; Liu, C.; Finkel, T. The role of ZKSCAN3 in the transcriptional regulation of autophagy. Autophagy. 2017;13(7):1235-1238.
  4. Liang, Y.; Choo, S.H.; Rossbach, M.; Baburajendran, N.; Palasingam, P.; Kolatkar, P.R. Crystal optimization and preliminary diffraction data analysis of the SCAN domain of Zfp206. Acta Crystallogr Sect F Struct Biol Cryst Commun. 2012;68(Pt 4):443-447.
  5. Groner, A.C.; Meylan, S.; Ciuffi, A.; Zangger, N.; Ambrosini, G.; Dénervaud, N.; Bucher, P.; Trono, D. KRAB-zinc finger proteins and KAP1 can mediate long-range transcriptional repression through heterochromatin spreading. PLoS Genet. 2010;6(3):e1000869.
  6. Rosspopoff, O.; Trono, D. Take a walk on the KRAB side. Trends Genet. 2023;39(11):844-857.
  7. Yang, L.; Hamilton, S.R.; Sood, A.; Kuwai, T.; Ellis, L.; Sanguino, A.; Lopez-Berestein, G.; Boyd, D.D. The previously undescribed ZKSCAN3 (ZNF306) is a novel “driver” of colorectal cancer progression. Cancer research. 2008;68(11):4321-4330.
  8. Urrutia, R. KRAB-containing zinc-finger repressor proteins. Genome Biol. 2003;4(10):231.
  9. Wang, W.; Cai, J.; Wu, Y.; Hu, L.; Chen, Z.; Hu, J.; Chen, Z.; Li, W.; Guo, M.; Huang, Z. Novel activity of KRAB domain that functions to reinforce nuclear localization of KRAB-containing zinc finger proteins by interacting with KAP1. Cell Mol Life Sci. 2013;70(20):3947-3958.
  10. Rosspopoff, O.; Trono, D. Take a walk on the KRAB side: (Trends in Genetics, 39:11 p:844-857, 2023). Trends Genet. 2024;40(2):203-205.
  11. Helleboid, P.Y.; Heusel, M.; Duc, J.; Piot, C.; Thorball, C.W.; Coluccio, A.; Pontis, J.; Imbeault, M.; Turelli, P.; Aebersold, R., et al. The interactome of KRAB zinc finger proteins reveals the evolutionary history of their functional diversification. Embo j. 2019;38(18):e101220.
  12. Han, Y.; Wu, P.; Wang, Z.; Zhang, Z.; Sun, S.; Liu, J.; Gong, S.; Gao, P.; Iwakuma, T.; Molina-Vila, M.A., et al. Ubiquinol-cytochrome C reductase core protein II promotes tumorigenesis by facilitating p53 degradation. EBioMedicine. 2019;40:92-105.
  13. Li, W.F.; Alfason, L.; Huang, C.; Tang, Y.; Qiu, L.; Miyagishi, M.; Wu, S.R.; Kasim, V. p52-ZER6: a determinant of tumor cell sensitivity to MDM2-p53 binding inhibitors. Acta Pharmacol Sin. 2023;44(3):647-660.
  14. Huang, C.; Wu, S.; Li, W.; Herkilini, A.; Miyagishi, M.; Zhao, H.; Kasim, V. Zinc-finger protein p52-ZER6 accelerates colorectal cancer cell proliferation and tumour progression through promoting p53 ubiquitination. EBioMedicine. 2019;48:248-263.
  15. Li, X.M.; Wen, J.H.; Feng, Z.S.; Wu, Y.S.; Li, D.Y.; Liang, S.; Wu, D.; Wu, H.L.; Li, S.M.; Ye, Z.N., et al. Effect of Lacking ZKSCAN3 on Autophagy, Lysosomal Biogenesis and Senescence. Int J Mol Sci. 2023;24(9).
  16. Cho, Y.E.; Kim, J.H.; Che, Y.H.; Kim, Y.J.; Sung, J.Y.; Kim, Y.W.; Choe, B.G.; Lee, S.; Park, J.H. Role of the WNT/β-catenin/ZKSCAN3 Pathway in Regulating Chromosomal Instability in Colon Cancer Cell lines and Tissues. Int J Mol Sci. 2022;23(16).
